# The Generalized Phase Rule, the Extended Definition of the Degree of Freedom, the Component Rule and the Seven Independent Non-Compositional State Variables: To the 150th Anniversary of the Phase Rule of Gibbs

**DOI:** 10.3390/ma17246048

**Published:** 2024-12-10

**Authors:** George Kaptay

**Affiliations:** 1Institute of Physical Metallurgy, Metal Forming and Nanotechnology, University of Miskolc, 3515 Miskolc, Hungary; kaptay@hotmail.com; Tel.: +36-30-415-0002; 2HUN-REN-ME Materials Science Research Group, University of Miskolc, 3515 Miskolc, Hungary

**Keywords:** components, state variables, phase rule, degree of freedom, constraints

## Abstract

The phase rule of Gibbs is one of the basic equations in phase equilibria. Although it has been with us for 150 years, discussions, interpretations and extensions have been published. Here, the following new content is provided: (i). the choice of independent components is discussed, and the component rule is introduced, (ii). independent state variables are divided into compositional and non-compositional ones, (iii). the generalized phase rule is derived replacing number two in the original phase rule by the number of independent non-compositional state variables introduced above, (iv). the degree of freedom is decreased by the number of compositional constraints in special points (azeotrope and congruent melting) of phase diagrams, (v). a rule is derived connecting the maximum number of coexisting phases with the dimensions of the phase diagram, (vi). examples show how to apply the phase rule to unary, binary and ternary phase diagrams and their sections, (vii). the same is extended with the discussion of calculable and not calculable phase fractions, (viii). it is shown that the current definition of the degree of freedom is not sufficient in the number of cases, (ix). the current definition of the degree of freedom is extended, (x). the application of the generalized phase rule is demonstrated when other non-compositional state variables are applied for nano-phase diagrams, and/or for phase diagrams under the influence of electric potential difference, external magnetic field, mechanical strain or the gravitational field.

## 1. Introduction

Chemical thermodynamics or equilibria of materials are essential chapters of materials science [1,2,3,4,5,6,7,8,9,10]. Here, let us summarize shortly how the calculations are performed (see also the schematic Figure 1):i.First, the engineer selects in an arbitrary way (upon his/her interest) the state parameters under which the calculations are performed; they are usually the components (and the species behind), their average molar (or mass) ratios, temperature and pressure; the engineer has technical ways to enforce all these quantities on the system.ii.Second, the engineer might select the experimental path; in this case, a long experiment should be run under the above selected values of the state parameters and at the end of the experiment (supposing the system is driven from its initial state to its equilibrium state by the laws of Nature), the characteristics of the equilibrium state should be measured as follows: the number and identity of the equilibrium phases that are formed, the equilibrium compositions of those phases and the equilibrium phase (or mass) fractions of those phases.iii.Alternatively, the engineer might select the theoretical path to calculate the same characteristics of the equilibrium state instead of measuring them; in this case, the engineer (a). should make a list of all possible phases, (b). should collect the partial molar Gibbs energy functions of all components in all phases preferably from an available databank (alternatively they can be measured or modeled), (c). should make the list of all possible phase combinations, (d). for each phase combination, the equilibrium phase compositions should be calculated by Equations (3a)–(4c) and the equilibrium phase fractions should be calculated by Equations (9a) and (9f); from here, the average molar Gibbs energy of the system should be found for each phase combination, (e). finally, the equilibrium phase combination should be selected by finding the most negative average molar Gibbs energy of the system. Phase equilibria can be understood better, and the list of phase combinations can be considerably shortened if the maximum number of coexisting phases is calculated by the phase rule discussed in this paper.

The phase rule of Gibbs is about the number of equilibrium phases in the system (*P*). Phases are 3D bodies that differ from each other by their inner structure and/or composition. If the same phase is broken into many parts (as many bubbles, droplets or crystals) it does not increase the number of phases, i.e., one million graphite crystals in the same crucible is one phase. The phase rule of Gibbs connects the maximum number of phases (*P_max_*) that can coexist with planar interfaces in an equilibrium with the number of independent components in the system (*C*) [1]:(1a)$$P_{max}=C+2$$
where “2” corresponds to pressure and temperature. It is important to note that both the original phase rule of Gibbs and its extension offered here are valid only in truly equilibrium systems, in which chemical reactions and phase changes are not limited by kinetic constraints. In real life, however, especially at relatively low temperatures, “non-reacting systems” are sometimes discussed, in which chemical reactions and/or phase changes seem not to happen due to our limited human lifetime. The phase rule of Gibbs should not be applied to such “non-reactive systems”, simply because those are not in real equilibrium. In fact, in those non-equilibrium, non-reacting systems, one can find as many phases as many are added by the engineer to the system, in seeming contradiction with Equation (1a).

Equation (1a) plays a major role not only in the interpretation of alloy phase diagrams [2,3,4], but also in their calculation by the Calphad method (Calphad = Calculation of Phase Diagrams) [5,6,7,8,9]. Equation (1a) is frequently written through the “degree of freedom” (*F*, dimensionless) or “variance”, defined as [10,11,12,13,14,15,16,17,18,19,20,21,22]:(1b)$$F\equiv P_{max}-P$$
where *P* is the actual number of phases in the system. The textual definition of the degree of freedom (*F*) widely used in the literature is as follows: “the degree of freedom is the number of those independent state variables whose values can be freely changed within a finite interval without the appearance or disappearance of any phase” [13]. In Section 9 of this paper, it will be shown that for certain important cases of phase equilibria, this definition is not sufficient.

Replacing $P_{max}$ in Equation (1b) by Equation (1a),
(1c)F=C+2−P

Replacing *P* by *P_max_* in Equation (1c), the limiting value of *F* = 0 is obtained meaning the system is non-variant. Thus, for the maximum number of phases to coexist in the system, all state variables must have their fixed values dictated by the laws of Nature. However, if $P<P_{max}$, then F>0 and then we have some freedom to select the values of *F* state variables as explained above and in Section 9.

In the literature, the phase rule is derived and interpreted in two ways. One of the ways considers potentials, such as chemical potentials of the components, temperature and pressure [8,9,13]. Taking into account the Gibbs–Duhem equation, the degree of freedom is obtained as the number of independent potentials. As a consequence, phase diagrams are also obtained as functions of the same independent potentials. However, there are two problems with this approach: (i) the chemical potentials of all components are temperature- and pressure-dependent, so it is hard to see how they can be independent potentials from pressure and temperature, (ii) the resulting phase diagrams are far from being practical: materials engineers or chemical engineers never use phase diagrams with chemical potentials on their axes, simply because there is no technical way to enforce their values on a system (note: there are technical ways to enforce the values of pressure and temperature on the system). For these two reasons, this approach will not be followed in this paper.

The second way is to consider state variables (see for example [3]), such as the average mole fractions of components in the system, pressure and temperature. Then, from comparing the number of independent equations on heterogeneous phase equilibria with the number of independent parameters determining the chemical potentials of the components that are made equal in the aforementioned equations, the same phase rule follows (see Section 4). Then, the degree of freedom is obtained as the number of independent state variables. As a consequence, phase diagrams are also obtained as functions of the same independent state variables: pressure, temperature and average mole fractions of components in the system. There are two advantages of this second method: (i) the average mole fractions of the components in a system, pressure and temperature are truly independent of each other, (ii) the resulting phase diagrams are practical, and so all materials/chemical engineers use phase diagrams with average mole fractions of components, temperature and pressure along their axes, as there are simple technical ways to enforce their values on the system. Thus, this second approach is followed in this paper. However, following this approach does not mean all the results will be identical to the previous results (see Section 4).

Another problem with the original phase rule is that in different situations, number “2” in Equations (1a)–(1c) is usually not valid. This might happen when the values of pressure and/or temperature are fixed (see Section 7), or if further independent non-compositional state variables are considered (see Section 11), or if compositional constraints are enforced (see Section 5). This problem is resolved in the literature by replacing “2” with other positive integer numbers (or zero) based on some additional discussions before Equations (1a) and (1c) are applied. However, such additional discussions (even if usually correct) are somewhat cumbersome after Equations (1a) and (1c) are declared to be “the phase rule”. To fix this problem, in this paper, an improved form of the phase rule is derived with the following properties: (i). the new equations reduce back to Equations (1a) and (1c) in the simplest case, (ii). the new equations can be applied without a need to modify them each time before they are applied (see Section 4). In this way, the improved phase rule will be more correct and it will also be easier to apply.

Summarizing, the following aspects of the phase rule are discussed in his paper:-in Section 2, the choice of components is discussed and the component rule is introduced;-in Section 3, independent state variables are divided into compositional and non-compositional ones;-in Section 4, the generalized phase rule is derived replacing number 2 in Equations (1a) and (1c) with the number of independent non-compositional state variables introduced above;-in Section 5, the degree of freedom is decreased by the number of compositional constraints in special points (azeotrope and congruent melting) of phase diagrams, but without decreasing the maximum number of coexisting phases in the same phase diagram;-in Section 6, a rule is derived connecting the maximum number of coexisting phases with the dimensions of the phase diagram, or their sections;-in Section 7, examples show how to apply the phase rule to unary, binary and ternary phase diagrams and their sections;-in Section 8, the same is extended with the discussion of calculable and not calculable phase fractions;-in Section 9, it is shown that the above definition of the degree of freedom is not sufficient in the cases of calculable phase fractions (except when it is unity);-in Section 10, the above definition of the degree of freedom is extended by claiming that the phase fractions of phases must not be changed upon changing the *F* number of state variables;-in Section 11, the application of the generalized phase rule is demonstrated when other non-compositional state variables are applied for nano-phase diagrams, and/or for phase diagrams under the influence of electric potential difference, external magnetic field, mechanical strain or the gravitational field.

## 2. Species of Interest, Components and the Component Rule

As Equations (1a) and (1c) contain the number of independent components (*C*), first this question is discussed here. Unfortunately, in some fields and in some simplified texts, expressions like elements, atoms, molecules, compounds, species and components are used as synonyms. So, before the number of components is defined here, let us show the differences between the meanings of these words.

First, let us explain the difference between elements and atoms. Atoms exist in Nature and their existence has been proven at least for a century. Although we know that atoms are made of elementary particles, nevertheless in materials science, atoms are convenient to consider as the smallest building blocks of materials (sometimes remembering how important electrons are in making chemical bonds, or in electrochemistry). For example, silicon atoms can diffuse and evaporate, can be part of a metallic alloy or can make chemical bonds with other atoms. On the other hand, the word “element” has an abstract meaning: silicon as an element and its symbol Si can be positioned into the Periodic System and it can be discussed. However, silicon as an element does not diffuse.

Molecules are made of atoms. Molecules are real entities, they can diffuse, dissociate, evaporate, etc. On the other hand, the word “compound” has an abstract meaning. We can talk about compounds, but they do not diffuse.

Both the words species and components have abstract meanings. The word species is used in the most general sense: they are all entities that can be described by a chemical formula unit, such as C, H, H_2_, CH, CH_4_, etc.… That is why in this paper I discuss “species of interest” to the engineer and to the researcher, meaning that any species can come to the mind of an engineer and he/she can be interested in any species. However, the word component is used in a narrower sense here: all components are species, but not all species are necessarily components. Components in this paper are treated in the thermodynamic sense: components are those chemically independent selected species which determine phase equilibria in a given system. That is why components play a crucial role in this paper, as the phase rule is about phase equilibria.

In metal sciences, components are selected as chemical elements. They are indeed independent of each other, as in materials technologies the atomic nuclei are not modified, and so chemical elements cannot be transformed into each other.

An additional problem arises, when in chemical, ceramic and polymer engineering, compounds are preferably selected for components instead of elements. In this case, one should first select a full list of species of interest and then the components should be selected from this list, usually neglecting some of the species of interest. In this case, the components should be chemically independent, i.e., no chemical reaction should exist between them. Sometimes the number of independent components is calculated as the number of species minus the number of independent reactions between them [3,4]. However, if the number of species is large (as for example in organic chemistry or polymers engineering), this is not an easy task.

To make it easier to find the maximum number of components, the following “component rule” is introduced here: “the maximum number of components to be selected for phase equilibria calculations from a list of species of interest (elements and compounds) equals the total number of elements contained in all those species of interest.” This can be proven as follows: all compounds of interest can be synthesized by reacting some elements in given proportions. So, it is sufficient to select the elements to define all the species of interest. Let us note that the component rule is identical with the idea written above: the number of independent components is the number of species minus the number of independent chemical reactions between them. However, it is easier to count the number of elements in the mixture of 100 hydrocarbons (=2: C and H) than counting the number of independent chemical reactions between them (=98).

The above rule does not mean that all or any of the selected components must be an element; in fact, some or all of them can be also compounds. However, in this case, care should be taken to select only components (i). with no chemical reactions between them, and (ii). with a composition range covering the full composition range of species of interest. Note: if all elements that make up the species of interest are also selected as species of interest, then only those elements can be selected as components to obey the above rule. To simplify the situation, a smaller number of components can also be selected than the maximum value given by the component rule, supposing the above rules are obeyed. However, in this case, usually the dissociation of compound-components cannot be studied.

Following the above rules, usually some species of interest are excluded from the list of components. However, after phase equilibrium is found as step 1 (discussed in this paper), the concentrations of all the species of interest can be calculated in all equilibrium phases as step 2, using the widely known methods of the thermodynamics of chemical reactions. Let us present some examples:

Example 1, for metallurgical engineers: suppose the species of interest are Al, Al_3_Ti, AlTi, AlTi_3_, Ti. Although there are five species, there are only two elements (Al and Ti), and so the maximum number of components is two. As both elements are included as species of interest, the only possible choice for the components is Al and Ti. Reacting them in different proportions, all other species of interest Al_3_Ti, AlTi and AlTi_3_ are obtained.

Example 2, for ceramic engineers: suppose the species of interest are FeO, Fe_3_O_4_ and Fe_2_O_3_. Although there are three species, there are only two elements (Fe and O), so the maximum number of components is two. A metallurgical engineer would select Fe and O as the two components. However, a ceramic engineer prefers oxides and should select FeO and Fe_2_O_3_ as the two components. This choice is perfect, as in reacting these two oxides, the third species of interest Fe_3_O_4_ can be obtained. Note that the choice of FeO and Fe_3_O_4_ as the two components is wrong, as their reaction cannot lead to the third species of interest Fe_2_O_3_, simply because the range of mole fractions of O in the selected FeO and Fe_3_O_4_ components (0.500 … 0.571) does not include the mole fraction of O (0.600) in the third species of interest (Fe_2_O_3_).

Example 3, for chemical engineers: suppose the species of interest are methane (CH_4_ with 0.800 mole fraction of H), ethane (C_2_H_6_, with 0.750 mole fraction of H) and benzene (C_20_H_42_ with 0.678 mole fraction of H). Although there are three species, there are only two elements (C and H), so a maximum of two components can be selected. One of the possibilities is to select carbon and H_2_, as in reacting them with each other, all the three species of interest can be created (at least in principle). But this is not a usual choice of a chemist. For a chemist, an ideal choice for the two components is CH_4_ and C_20_H_42_, as by combining them in a given ratio, the third species of interest can be created (at least in principle). This is because the mole fraction range of H in these two components (0.678 … 0.800) covers the mole fractions of H in C_2_H_6_ (0.750). Note that the choice of CH_4_ and C_2_H_6_ (as the mixture of the two simplest species from the above list) as two components would be wrong, as the third species of interest cannot be created from their combination, even in principle. It is because the mole fraction range of H in these two components (0.750 … 0.800) excludes the mole fractions of H in C_20_H_42_ (0.678).

## 3. The Four Ways to Classify the State Parameters

State parameters (*SP*s) play a major role in the development of the phase rule, so let us discuss them here. First, let us define them: *SP*s are those independent physical/chemical quantities (i). that have significant influence on the studied phase equilibria through their influence on the molar Gibbs energy of at least one phase in the system, and (ii). whose values can be freely selected by us, as we have technical ways to force their values on the system. That is why the average mole fractions of the components are *SP*s, but their chemical potentials are not. The *SP*s can be classified in the following four ways.

According to the first way of classification, *SP*s can be variables or can have constant values. As only variable *SP*s have any influence on the phase rule, they are called here as “state variables”, abbreviated as *SV*s. Note that all *SV*s are *SP*s, but not all *SP*s are necessarily *SV*s, as some of the *SP*s can be selected by us to have constant values during constructing certain types of phase diagrams or their sections.

According to the second way of classification, *SV*s can be independent or dependent. As only independent *SV*s have any independent influence on the phase rule, they are called here as “independent state variables”, abbreviated as *ISV*s, and their number is denoted as *N_ISV_* (dimensionless).

According to the third way of classification, *ISV*s can be compositional or non-compositional. The compositional *ISV*s are the independent average mole fractions of the components in the system (*x_i_*). Their number is (*C* − 1), as the *x_i_* value of the *C*th component follows from the balance equation $\sum_{i}x_{i}=1$. However, this number is decreased by each compositional constraint enforced between the compositional state variables (such as x_A_/x_C_ = 0.5) to obtain the real number of independent compositional state variables. The number of these compositional constraints is denoted as Z_C_ (dimensionless). As independent compositional state variables cannot have a negative value: $Z_{C}\leq C-1$. As follows, $Z_{C}$ might have a positive value only for binary and multi-component systems, and its value depends upon our decision.

The non-compositional *ISV*s are classically pressure (*p*) and temperature (*T*), although other quantities also exist (see Section 11). Let us abbreviate them as *NC-ISV*s and let us denote their number as *N_NC-ISV_*. This number is decreased by each non-compositional constraint applied between them (such as *p*/*T* = constant) to obtain the real number of independent non-compositional state variables. The number of non-compositional constraints is denoted as *Z_NC_* (dimensionless). As their number cannot have a negative value: $Z_{NC}\leq N_{NC-ISV}$. The value of $Z_{NC}$ depends on our decision.

Summarizing: the total number of independent state variables is as follows:(2a)$$N_{ISV}=C-1-Z_{C}+N_{NC-ISV}-Z_{NC}$$

According to the fourth way of classification, *ISV*s can be the following: (a) *ISV*s of free values are called here the (number of) “degrees of freedom”, denoted as *F* and defined by Equation (1b), (b) *ISV*s of fixed values with their number denoted as $N_{ISV-FIX}$. The values of *ISV*s of fixed values are fixed by the laws of Nature to ensure the equilibrium of given phases in given regions of phase diagrams (see also Section 9). In contrast, the values of *ISVs* of free values can be selected freely in a finite interval, i.e., they are not fixed by the laws of Nature to ensure equilibrium of given phases in given regions of phase diagrams (see also Section 9). Obviously, their sum can be written as follows:(2b)$$N_{ISV}=F+N_{ISV-FIX}$$

## 4. The General Derivation of the Phase Rule

In this section, a well-known method of derivation is repeated, but the result will be novel, thanks to the new types of state variables introduced in the previous section.

The phase rule of Gibbs follows from his condition for heterogeneous equilibria [1]. For a two-component (*A-B*) and a two-phase (α, β) system, this condition is written as follows:(3a)$$G_{m,A\left(\alpha \right)}=G_{m,A(\beta )}$$
(3b)$$G_{m,B\left(\alpha \right)}=G_{m,B(\beta )}$$
where $G_{m,i\left(\Phi \right)}$ (J/mol) is the partial molar Gibbs energy (or chemical potential) of component *i* (*i = A* or *B*) in phase Φ (Φ = *α* or β). Generalizing Equations (3a) and (3b), one can claim that the condition of heterogeneous equilibrium in any system is the equality of the partial molar Gibbs energies of each component in all phases and also in the whole system, including the interfaces. For equilibrium, this condition should be true for all components.

Now, let us count how many independent Equations (3a) and (3b) can be written: their number is denoted here as *EQ* (dimensionless). First, let us claim that Equation (3a) is independent of Equation (3b), i.e., the fact that Equation (3a) is obeyed does not necessarily mean that Equation (3b) is also obeyed. This is because the value of $G_{m,A\left(\alpha \right)}$ in a given phase does not determine the value of $G_{m,B\left(\alpha \right)}$ in the same phase and vice versa. Note: the Gibbs–Duhem equation [3,4,5,6,7,8,9,23,24,25,26,27,28] connects only the partial derivatives of partial molar Gibbs energies of the components in each phase, but not their absolute values. Thus, it follows that the number of independent equations of type (3a) and (3b) is proportional to the number of independent components: EQ~C.

In addition to the above proportionality, *EQ* will obviously depend also on the number of equilibrium phases in the system (*P*). To analyze this relationship, let us consider the conditions of equilibrium in a one-component (*A*), three-phase (α-β-γ) system:(4a)$$G_{m,A\left(\alpha \right)}=G_{m,A(\beta )}$$
(4b)$$G_{m,A\left(\beta \right)}=G_{m,A(\gamma )}$$
(4c)$$G_{m,A\left(\alpha \right)}=G_{m,A(\gamma )}$$

It is obvious that if Equations (4a) and (4b) are obeyed, then Equation (4c) will be automatically obeyed. This is because if α = β and β = γ, then obviously also α = γ. Then, only two of three Equations (4c) are independent. Generalizing this observation, (*P* − 1) independent equations of type (4a)–(4c) are sufficient to guarantee heterogeneous equilibrium in a 1-component *P*-phase system. Therefore, heterogeneous equilibrium in a *C*-component *P*-phase system can be guaranteed by the following number of independent equations of type (3a)–(4c):(5a)EQ=C·(P−1)

Now, let us find the number of parameters that determine the values of the partial molar Gibbs energies of Equations (3a)–(4c):$\;G_{m,i(\Phi )}=f(x_{i\left(\Phi \right),}T,\;p,\;\ldots )$, where the three points mean other possible independent non-compositional state variables. For each phase Φ, there are C−1 independent mole fractions $x_{i\left(\Phi \right)}$, as the mole fraction of the $C^{th}$ component in this phase is found by the balance equation $\sum_{i}x_{i(\Phi )}=1$. Then, for a *C*-component and *P*-phase system, the number of independent parameters (*PAR*, dimensionless) in model equations $G_{m,i(\Phi )}=f(x_{i\left(\Phi \right),}T,\;p,\;\ldots )$ can be written as follows:(5b)$$PAR=P\cdot \left(C-1\right)+N_{NC-ISV}-Z_{NC}$$

If the parameter values are searched under the conditions of a particular heterogeneous equilibrium, then the values of one or more *PAR*-s become unknowns: for example, the melting temperature of a phase is an unknown temperature if the condition for the coexistence of solid and liquid phases is studied. Mathematically, the situation is obvious when *EQ = PAR*, i.e., when the number of independent equations on heterogeneous phase equilibria is the same as the number of independent unknowns in the model equations for the partial molar Gibbs energies. This is the case for nonvariant phase equilibria, when all the state parameters have fixed values dictated by the laws of Nature, and therefore the number of equilibrium phases has its maximum value. Thus, the value of *P_max_* follows by substituting Equations (5a) and (5b) into the equality *EQ* = *PAR*:(5c)$$P_{max}=C+N_{NC-ISV}-Z_{NC}$$

One can see that Equation (5c) is identical with Equation (1a) for the classical case of $N_{NC-ISV}-Z_{NC}$ = 2 (*p* and *T*). Now, let us provide some additional proof that the number of phases expressed from the equality of Equations (5a) and (5b) is indeed the maximum number of coexisting phases. To prove this, let us consider the following two cases:

Case 1. Suppose *EQ* = *PAR* + 1, i.e., more equations than unknowns: substituting Equations (5a) and (5b) into this equality, the following equation is obtained: $P_{1}=C+N_{NC-ISV}-Z_{NC}+1$, comparing this equation with Equation (5c): $P_{1}>P_{max}$. To judge its meaning, let us give a simple example for the case of more equations than unknowns: *x* = 1 and *x* = 2. One can see that there is no mathematical solution for this system of equations, as *x* cannot equal to 1 and to 2 at the same time. Therefore, this case cannot exist in Nature, i.e., the case $P>P_{max}$ is excluded. By excluding this case, the “*max*” subscript in Equation (5c) is partly proven.

Case 2. Suppose *EQ = PAR* − 1, i.e., more unknowns than equations: substituting Equations (5a) and (5b) into this equality, the following equation is obtained: $P_{2}=C+N_{NC-ISV}-Z_{NC}-1$, comparing this equation with Equation (5c): $P_{2}<P_{max}$. To judge its meaning, let us give a simple example for this case of fewer equations than unknowns: *x + y* = 3. This equation is obeyed by an infinite number of mathematical solutions: one of them is *x* = 1 and *y* = 2, another one is *x* = 2 and *y* = 1, etc. Thus, this case of $P<P_{max}$ is possible, proving further the validity of the subscript “max” in Equation (5c). Extending the logic shown here, the degree of freedom is written as follows:(6a)F=PAR−EQ

Substituting Equations (5a) and (5b) into Equation (6a),
(6b)$$F=C+N_{NC-ISV}-Z_{NC}-P$$

One can see that Equation (6b) is identical with Equation (1c) in the simplified case of $N_{NC-ISV}-Z_{NC}$ = 2 (*p* and *T*). Let us replace $C+N_{NC-ISV}-Z_{NC}$ in Equation (6b) by $P_{max}$ in agreement with Equation (5c): then, Equation (1b) is obtained, i.e., Equations (1b) and (6a) are identical equations to express the degree of freedom.

Substituting Equations (2a) and (6b) into Equation (2b), the expression for the number of non-compositional state variables of fixed values by the laws of Nature is obtained as follows:(6c)$$N_{ISV-FIX}=P-1-Z_{C}$$

Equations (2a), (5c) and (6a)–(6c) together form the essence of the generalized phase rule except the case of special points in phase diagrams (see in the next section). Let us note that according to Equations (5c) and (6b), the values of the most essential parameters of the phase rule (*P_max_* and *F*) are not dependent on $Z_{C}$, i.e., on the number of constraints enforced between the compositional state variables. However, both of them depend on $Z_{NC}$, i.e., on the number of constraints enforced between the non-compositional state variables. The fact that constraints enforced between non-compositional and compositional state variables affect differently the major parameters of the phase rule is sufficient to explain why the compositional and non-compositional state variables should have been separated in the previous section.

As follows from the above derivation, the phase rule is valid only for those phases that are in equilibrium with each other at equilibrium values of *ISV*s under the condition of Equations (3a)–(4c). As was shown by Laughlin [29], this is the case only for phases that form from each other by first-order phase transitions.

Let us note that Equations (2a), (5c) and (6a)–(6c) are more advanced compared to the classical Equations (1a) and (1c), as number 2 in Equations (1a) and (1c) is now replaced by the expression $N_{NC-ISV}-Z_{NC}$, i.e., by the number of independent non-compositional state variables, which can have a value of 2, or can have other values such as 0 or even values larger than 2, depending on our decision (see also Section 11). It means that Equations (2a), (5c) and (6a)–(6c) can be used as they are, without a need to change them each time before they are applied, as is the case with Equations (1a) and (1c). Instead, as is the case with all reasonable equations in natural sciences (except Equations (1a) and (1c)), only appropriate parameter values should be substituted into them without changing those equations each time before they are applied.

## 5. The Role of Compositional Constraints Valid in Special Points of Some Phase Diagrams

In certain phase diagrams, there are special points of equal compositions of different phases at the same temperature and pressure: $x_{B(\alpha )}=x_{B(\beta )}$. Examples are (a) the azeotrope point and (b) the congruent melting point of a compound phase. The number of these constraints is denoted as $Z_{P}$ (dimensionless). As this constraint is true only for a single point of the phase diagram, it does not affect the maximum number of phases in the whole phase diagram written by Equation (5c): $P_{max}=C+N_{NC-ISV}-Z_{NC}$. It also does not affect the degree of freedom written by Equation (6b): $F=C+N_{NC-ISV}-Z_{NC}-P$, at least in all points of the phase diagram except the special points. The value of $Z_{P}$ only affects the number of parameters (*PAR*) in the special points of the phase diagrams. Indeed, the number of independent parameters in functions $G_{m,i(\Phi )}=f(p,T,x_{i\left(\Phi \right)},\;\ldots )$ is decreased by $Z_{P}$ compared to Equation (5b), if constraints of type $x_{B(\alpha )}=x_{B(\beta )}$ are applied:(7a)$$PAR^{*}=P\cdot \left(C-1\right)+N_{NC-ISV}-Z_{NC}-Z_{P}$$

Now, let us express the degree of freedom valid in the special points (denoted as $F^{*}$), by substituting Equations (5a) and (7a) into Equation (6a):(7b)$$F^{*}=C-P+N_{NC-ISV}-Z_{NC}-Z_{P}$$

In the simplest case of $N_{NC-ISV}-Z_{NC}-Z_{P}$ = 2, Equation (7b) simplifies back to Equation (1c). Note that a similar equation ($F^{*}=C-P+2-Z_{P}$ using our notations) was derived by Hillert [13] (see his Equation (12)). Note, however, that Hillert did not consider different possible numbers of independent non-compositional state variables, so our Equation (7b) is more general compared to Equation (12) of Hillert [13].

Now, let us substitute Equations (2a) and (7b) into Equation (2b) to express the number of state variables of fixed values by the laws of Nature in the special points, denoted as $N_{SV-FIX}^{*}$:(7c)$$N_{ISV-FIX}^{*}=P-1-Z_{C}+Z_{P}$$

Note that the values of $Z_{P}$ are compensated in the sum of Equations (7b) and (7c) to provide Equation (2a), being independent of $Z_{P}.$ The maximum possible value of $Z_{P}$ can be found by requesting that $F^{*}\geq 0$. Then, from Equation (7b),
(7d)$$Z_{P}\leq C-P+N_{NC-ISV}-Z_{NC}$$

## 6. A Rule for 2D Phase Diagrams

Now, let us compare Equations (2a) and (5c). As their right-hand sides contain the same expression $C+N_{NC-ISV}-Z_{NC}$, the following new relationship is established between them:(8a)$$P_{max}=N_{ISV}+1+Z_{C}$$

Let us consider a reduced case with $Z_{C}$ = 0 and let us denote the reduced maximum number of coexisting phases as $P_{max-red}$. Substituting $Z_{C}$ = 0 into Equation (8a),
(8b)$$P_{max-red}=N_{ISV}+1$$

Usually, phase diagrams are presented on 2D pages of journals or books (although in principle one can also present 3D diagrams on 2D pages, but their readability is questionable). Such diagrams on pages usually have two axes, along which two independent state variables are plotted (such as *p* and *T*, as an example). Substituting $N_{ISV}$ = 2 into Equation (8b), *P_max-red_* = 3. Thus, from this point of view, all 2D phase diagrams with $Z_{C}$ = 0 are similar. The three examples presented in the next section (see Figure 2, Figure 3 and Figure 4 and Table 1) provide three different phase diagrams, which look different, but are similar in a sense that for all of them $N_{ISV}$ = 2 and *P_max-red_* = 3 (also note that *F* = −1 for *P* = 4, meaning that four phases cannot coexist in equilibrium).

## 7. Some Examples on How to Apply the Improved Phase Rule

In Figure 2, Figure 3 and Figure 4, widely known examples of 1-2-3 component phase diagrams or their sections are shown. They are analyzed in Table 1. All the results given in Table 1 are widely known and agree with the equations derived above. All Figure 2, Figure 3 and Figure 4 agree with Equation (8b).

Note that although there are four phases in Figure 2 and Figure 3, all of them cannot coexist at the same time in equilibrium. Note also that in the special (congruent melting) point of Figure 3, the degree of freedom is reduced due to identical equilibrium compositions of the gamma and liquid phases, in accordance with Equation (7b).

Note also, that by increasing temperature, a special temperature can be found when the three-phase region in the middle of Figure 4 is concentrated into a single point, at which also a liquid phase appears: this point is called the ternary eutectic point with four co-existing phases (α + β + γ + *l*). This point can be shown in a 3D phase diagram as the function of concentrations and temperature (at fixed pressure). Similarly, a four-phase point can be shown as the function of compositions and pressure (at fixed temperature) in a 3D phase diagram [30]. The horizontal cross section at this ternary eutectic temperature is not shown in Figure 4, as the probability of finding this temperature is practically nil, i.e., four phases cannot be experimentally found to coexist in ternary systems with fixed pressure and temperature values. This also follows from Table 1, predicting *F* = −1 for this case, further proving that this case does not exist, in agreement with Equation (6b) shown above. In the same way, we can conclude that four phases cannot coexist in binary systems with fixed pressure and variable temperature or fixed temperature and variable pressure. For the same reason, four phases cannot coexist in equilibrium in one-component systems, even if both pressure and temperature are variables. This is because in all these cases, *P_max_* = 3 and *F* = −1 for the case of *P* = 4. This argument was written by Gibbs as follows: “it is entirely improbable that there are four coexistent phases of any simple substance” [1].

**Figure 2 materials-17-06048-f002:**
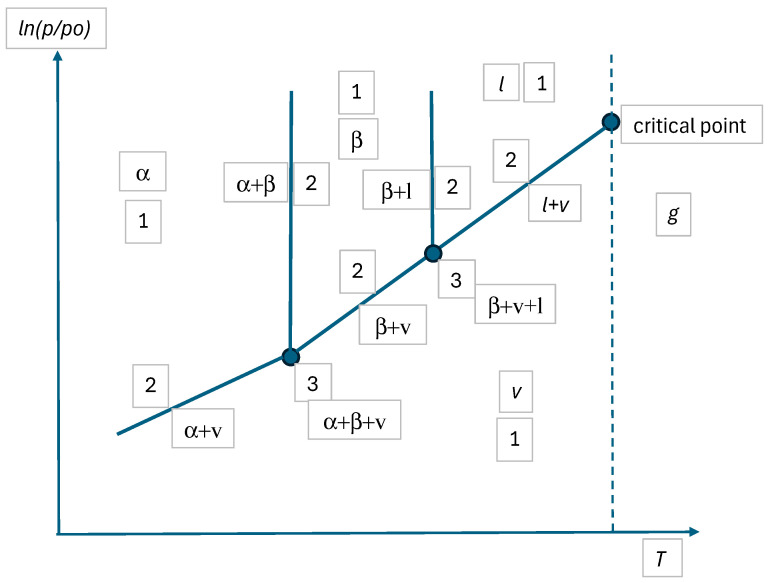
Schematic of a 1-component phase diagram with two allotropic phases. Numbers denote the number of coexisting, equilibrium phases (see Table 1). Symbols α and β denote different solid crystal types (allotropes), *l* = liquid, *v* = vapor, *g* = gas.

**Figure 3 materials-17-06048-f003:**
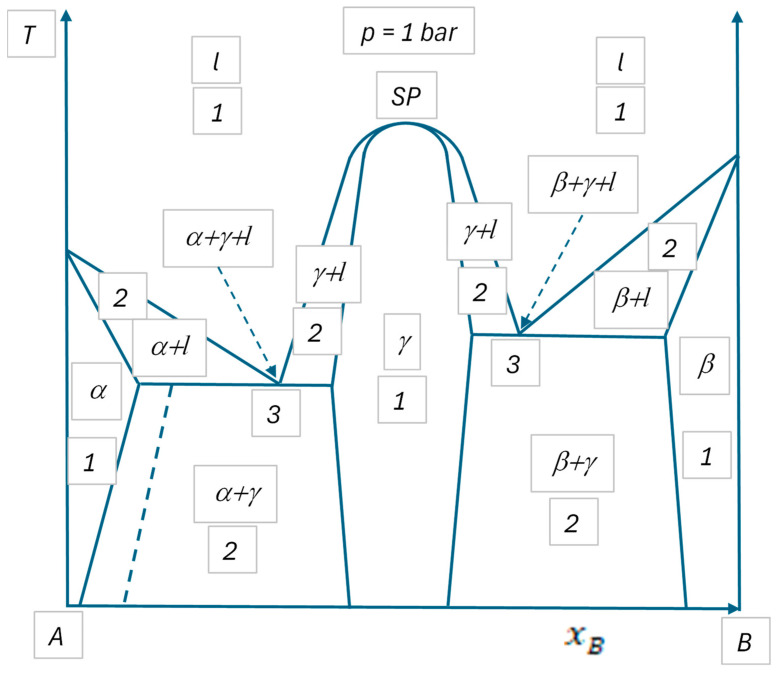
Schematic of the cross section at *p* = 1 bar of a 2-component phase diagram with two eutectic points and one congruently melting compound phase. Numbers denote the numbers of coexisting, equilibrium phases (see Table 1). Symbols α, β and γ denote different solid crystal types (the latter is a compound phase), *l* = liquid. “SP” is a special point = the congruent melting point of the compound phase with equal equilibrium compositions of the liquid phase and the compound phase γ. The dotted line in the 2-phase region α + γ corresponds to the constant phase fractions of $y_{\alpha }$ = 0.83, $y_{\gamma }$ = 0.17.

**Figure 4 materials-17-06048-f004:**
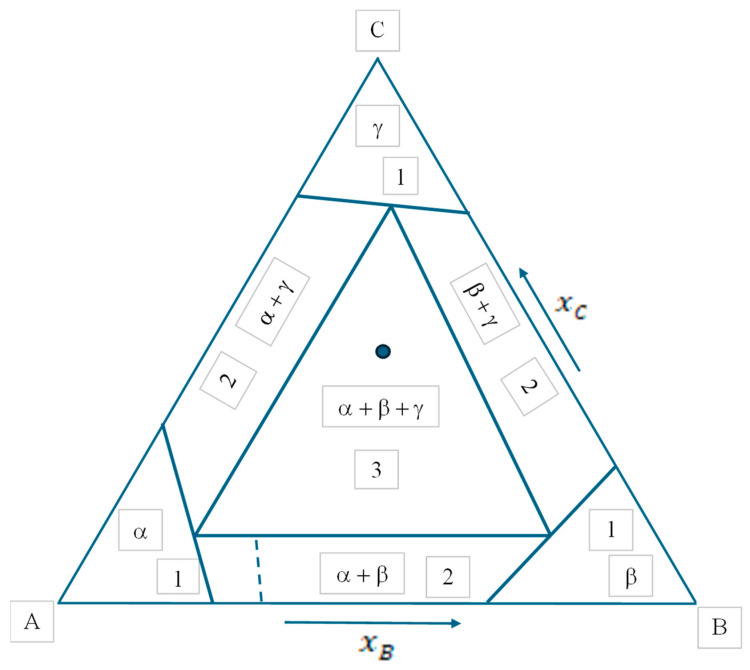
Schematic of the cross section at *p* = 1 bar and at a temperature below the ternary eutectic point of a 3-component phase diagram with three binary eutectic systems. Numbers denote the numbers of coexisting equilibrium phases (see Table 1). Symbols α, β and γ denote different solid crystal types. The dotted line in the 2-phase region α + β corresponds to the constant phase fractions of $y_{\alpha }$ = 0.83, $y_{\beta }$ = 0.17. The point in the 3-phase region α + β + γ corresponds to the constant phase fractions of $y_{\alpha }$ = 0.22, $y_{\beta }$ = 0.22, $y_{\gamma }$ = 0.56.

**Table 1 materials-17-06048-t001:** Applying the improved phase rule to phase diagrams shown in Figure 1, Figure 2 and Figure 3. *P_max_* is calculated by Equation (5c), *P* is taken as positive numbers up to *P_max_*, *F* and *N_ISV-FIX_* are calculated by Equations (6b) and (6c), except for the special point, where Equations (7b) and (7c) are applied. Equation (8b) is valid between *P_max_* and *N_ISV_*. Phase fractions are discussed in the next section.

Figure	*C*	*N_ISV_*	*P_max_*	*N_NC-ISV_ − Z_NC_*	*P*	*F*	*N_ISV-FIX_*	Phase Fractions
2	1	2 = *p*, *T*	3	2 = *p*, *T*	1	2	0	1
					2	1	1	not calculable
					3	0	2	not calculable
3	2	2 = *T*, *x_B_*	3	1 = *T*	1	2	0	1
					2	1	1	calculable
					3	0	2	not calculable
					2 (SP)	0	2	not calculable
4	3	2 = *x_B_*, *x_C_*	3	0	1	2	0	1
					2	1	1	calculable
					3	0	2	calculable
					(4)	(−1)	(3)	does not exist

Comparing Equations (8a) and (8b), one can conclude that 2D quasi-binary vertical sections of ternary or multicomponent phase diagrams (with $Z_{C}>$ 0) usually include points or regions with larger numbers of coexisting phases compared to 2D phase diagrams. As an example, let us consider a three-component system with $Z_{C}$ = 1 constraint between the average mole fractions of the components, such as *x_A_*/*x_C_* = 0.5. Then, a 2D vertical cross section of this ternary phase diagram can be plotted at *p* = 1 bar showing *T* and *x_B_* along its two axes. Substituting $N_{NC-ISV}-Z_{NC}$ = 1 (*T*) into Equation (5c), $P_{max}=C+1$ = 3 + 1 = 4 is obtained for this case. Now, Equation (8a) is valid instead of Equation (8b), providing the same result: $P_{max}$ = 2 + 1 + 1 = 4. To obtain a point with the four coexisting phases in a quasi-binary vertical section of a ternary eutectic phase diagram, a special value of the *x_A_*/*x_C_* ratio should be selected such that the line from the A–C binary side drawn towards the B-corner of Figure 4 passes through the ternary eutectic point with four equilibrium phases. However, selecting any other *x_A_*/*x_C_* ratio, the similar line in Figure 4 will miss the ternary eutectic point and so all other quasi-binary vertical sections of the same ternary eutectic phase diagram will not contain a point with four coexisting phases.

## 8. The Calculable and Not Calculable Phase Fractions

Equilibrium phase fractions have not been discussed in this paper so far, as they were not needed to derive the generalized phase rule. However, they are needed to explain the last column of Table 1 and to extend the definition of the degrees of freedom. The conditions for heterogeneous equilibria of Gibbs (Equations (3a)–(4c)) do not contain the phase fractions, and so the equilibrium phase fractions cannot be calculated from Equations (3a)–(4c). The equilibrium phase fractions are routinely calculated from the so-called materials balance equation:(9a)$$x_{B}=\sum_{\Phi }y_{\Phi }\cdot x_{B(\Phi )}$$
where $y_{\Phi }$ (dimensionless) is the equilibrium phase fraction of phase Φ of the system. For all phases in a system, an additional balance equation is also valid: $\sum_{\Phi }y_{\Phi }$ = 1. For a system containing a single phase (*P* = 1), the composition of this phase is obviously the same as the average composition of the system: $x_{B(\Phi )}=x_{B}$. Substituting this into Equation (9a), $y_{\Phi }$ = 1 is obtained at *P* = 1. That is why numbers 1-1 are shown in the last column of Table 1 in each row with *P* = 1.

Now, let us study multi-phase cases with P > 1. First, let us find the number of materials balance equations (9a), denoted as *EQ_MB_* (dimensionless). Equation (9a) is valid for each component in a *C*-component system, but the same equation would be redundant for the *C*th component, so the number of materials balance equations in a *C*-component and *P*-phase system is as follows:(9b)$${EQ}_{MB}=C-1$$

Now, let us calculate, how many independent phase fractions we have in a *C*-component, *P*-phase system. Because the phase fraction for the *P*th phase follows from the above-mentioned balance equation ($\sum_{\Phi }y_{\Phi }$ = 1), the number of independent phase fractions is *P* − 1. If this number equals or is smaller than the number of materials balance equations written by Equation (9b), then all the phase fractions can be calculated. That is why “calculable” is given in the last column of Table 1 for the following *C*/*P* cases: 2/2, 3/2, 3/3.

However, if P>C, then the number of materials balance equations written by Equation (9b) is not sufficient to find the phase fractions of all phases. That is why “not-calculable” is written in the last column of Table 1 for the following C/P cases: 1/2, 1/3, 2/3. Let us note that in these cases, we have an additional freedom to adjust the phase fractions by heating/cooling the system. Let us also note that the phase fractions would be calculable from measured enthalpy or volume of the system, but they are not considered in this paper as they are not independent state variables.

We have a special case for the special point “SP” of Figure 2, characterized by the compositional constraint $x_{B(\gamma )}=x_{B(l)}$ for the case of *P* = 2. For this case, the equilibrium mole fractions of the phases also equal the average composition of the system: $x_{B}=x_{B(\gamma )}=x_{B(l)}$. Substituting this into Equation (9a), the obvious equation $\sum_{\Phi }y_{\Phi }$ = 1 is obtained, from which the phase fractions of the particular phases cannot be calculated. That is why “not calculable” is written in the last column of Table 1 for the case of the special point denoted as “SP”.

## 9. Contradictions Identified Due to the Current Definition of the Degrees of Freedom

Let us first remind the common definition of the degree of freedom from Section 1: “the degree of freedom is the number of those independent state variables whose values can be freely changed within a finite interval without the appearance or disappearance of any phase” [13]. In this section, it will be shown that although this definition is sufficient to interpret simple cases, it is not sufficient to interpret more complex cases in analyzing different regions in different phase diagrams. To show this, let us first analyze different regions in Figure 2, for which the above definition is sufficient (probably this is the reason why in some texts, only this case is discussed in details):-Regions with *P* = 1, *F* = 2 (see Table 1): it means that both *T* and *p* can have any values in a finite interval independent of each other, without the appearance or disappearance of any phase. As the one-phase regions are 2D areas in Figure 2, logically *F* = 2 and so the phase rule and the above definition are in agreement.-Regions with *P* = 2, *F* = 1 (see Table 1): it means that only one of *T* or *p* can have any value in a finite interval and the second state variable has a fixed value by the laws of Nature to guarantee that no phase appears or disappears. As the two-phase regions are 1D lines in Figure 2, logically *F* = 1 and so the phase rule and the above definition are in agreement.-Regions with *P* = 3, *F* = 0 (see Table 1): it means that neither *T* nor *p* can have any freely selected value; rather, both should have fixed values by the laws of Nature to guarantee that no phase appears or disappears. As the three-phase regions are 0D points in Figure 2, logically *F* = 0 and so the phase rule and the above definition are in agreement.

Now, let us analyze different regions in Figure 3, for which the above definition is sufficient only in two regions out of the three:-Regions with *P* = 1, *F* = 2 (see Table 1): it means that both *T* and *x_B_* can have any value in a finite interval independent of each other, without the appearance or disappearance of any phase. As the one-phase regions are 2D areas in Figure 3, logically *F* = 2 and so the phase rule and the above definition are in agreement.-Regions with *P* = 2, *F* = 1 (see Table 1): it means that the value of only *T* or *x_B_* can be freely selected in a finite interval without the appearance or disappearance of any phase, while the second state variable must have a fixed value dictated by the laws of Nature. However, as the two-phase regions are 2D areas in Figure 3, logically *F* = 2 should be the case, and so the phase rule and the above definition of the degree of freedom are in contradiction.-Regions with *P* = 3, *F* = 0 (see Table 1): it means that neither *T* nor *x_B_* can have free values; rather, both of them must have predetermined values by the laws of Nature to guarantee that no phase appears or disappears. As the three-phase regions are 0D points in Figure 3, logically *F* = 0 and so the phase rule and the above definition are in agreement.

Now, let us analyze different regions in Figure 4, for which the above definition is sufficient only in one region out of the three:-Regions with *P* = 1, *F* = 2 (see Table 1): it means that both *x_B_* and *x_C_* can have any value in a finite interval independent of each other, without the appearance or disappearance of any phase. As the one-phase regions are 2D areas in Figure 4, logically *F* = 2 and so the phase rule and the above definition are in agreement.-Regions with *P* = 2, *F* = 1 (see Table 1): it means that the value of only *x_B_* or *x_C_* can be freely selected in a finite interval without the appearance or disappearance of any phase, while the second state variable must have a fixed value dictated by the laws of Nature. However, as the two-phase regions are 2D areas in Figure 4, logically *F* = 2 should be the case, and so the phase rule and the above definition of the degree of freedom are in contradiction.-Regions with *P* = 3, *F* = 0 (see Table 1): it means that neither *x_B_* nor *x_C_* can have free values; rather, both should have predetermined values by the laws of Nature to guarantee that no phase appears or disappears. However, as the three-phase region is a 2D area in Figure 4, logically *F* = 2 and so the phase rule and the above definition of the degree of freedom are in contradiction.

Let us conclude that contradictions arise only in those cases, in which the phase fractions are calculable and have different values from unity (see Table 1). In the next section, an extended definition of the degree of freedom is shown that helps to resolve the above identified contradictions without creating new ones.

Before doing so, let us mention that in some texts, the contradiction found for the above case of two-component, two-phase regions is wrongly “resolved” by claiming that the degree of freedom is 1, as at a freely selected temperature the equilibrium compositions of the equilibrium phases are fixed at the two ends of the tie-line and so the second degree of freedom has a fixed value (see [12], as an example). This reasoning is incorrect, because the degree of freedom should be selected from the list of independent state variables, but the equilibrium compositions of equilibrium phases are not in the list of state variables (note: the average mole fractions of the system are the real state variables—see Figure 1).

## 10. The Extended Definition of the Degree of Freedom

Now, let us search for a way out from the above-mentioned contradictions. Suppose the phase rule itself is correct, and the above contradictions appear only because the definition of the degree of freedom is not sufficient. This definition can be extended by an additional condition as follows: the degree of freedom (*F*) is the number of independent state variables that can be freely varied in a finite interval without the appearance or disappearance of any phase and without changing the selected value of the *X* physical quantity. In this extended definition, the latter part is the extension. In the problematic regions of Figure 3 and Figure 4, there are only two types of physical quantities *X* that can be measured as the function of the state variables: the equilibrium mole fractions of the components in the equilibrium phases ($x_{B(\Phi )}$) and the equilibrium phase fractions of the phases ($y_{\Phi }$). Let us consider them one by one:

First, let us suppose that *X* = $x_{B(\Phi )}$. To check if this is the case, let us consider a point selected in a two-phase region of Figure 3. This point is characterized by a tie-line, leading to two $x_{B(\Phi )}$ values of the two equilibrium phases along the two ends of the tie-line. Now, let us select another *T* as a free choice of a state variable of free value: then, we should find such an $x_{B}$ value, dictated by the laws of Nature, which would compensate the effect of changing the value of *T* in the values of $x_{B(\Phi )}$-s. When *T* is changed, the tie-line is shifted within the two-phase region, leading to different $x_{B(\Phi )}$ values. Unfortunately, changing the value of $x_{B}$ at a constant temperature along the new tie-line has no influence on the values of $x_{B(\Phi )}$, and so the originally selected values of $x_{B(\Phi )}$ cannot be obtained back. Thus, we can conclude that $X\neq x_{B(\Phi )}$.

Based on the above, we are left with only one remaining option: $X=y_{\Phi }$. This choice is also confirmed by the fact that contradictions are found above only for cases when the phase fractions are calculable and have different values from unity. Therefore, the extended definition of the degree of freedom is as follows: “the degree of freedom (*F*) is the number of independent state variables that can be freely varied in a finite interval without the appearance or disappearance of any phase and without changing the selected values of the equilibrium phase fractions of equilibrium phases”. To prove that the last part of this extended definition really helps to fix the above contradictory cases, let us discuss those cases again:

-Regions with *P* = 2 in Figure 3 and *F* = 1 (see Table 1): it means that the value of only *T* or *x_B_* can be freely selected, while the second state variable must have a fixed value dictated by the laws of Nature. This would logically lead to a 1D line within the two-phase regions. Indeed, constant values of the phase fractions in two-phase regions of Figure 3 are guaranteed along a line: see the dotted line in the two-phase region α+γ of Figure 3. Thus, the new definition is confirmed.-Regions with *P* = 2 in Figure 4 and *F* = 1 (see Table 1): it means that the value of only *x_B_* or *x_C_* can be freely selected, while the second state variable must have a fixed value dictated by the laws of Nature. This would logically lead to a 1D line in the two-phase regions of Figure 4. Indeed, constant values of the phase fractions in two-phase regions of Figure 4 are guaranteed along a line: see the dotted line in the two-phase region α+β of Figure 4. Thus, the new definition is confirmed.-A region with *P* = 3 in Figure 4 and *F* = 0 (see Table 1): it means that none of the values of *x_B_* and *x_C_* can be freely selected; rather, both should have fixed values dictated by the laws of Nature. This would logically lead to a 0D point in the three-phase region of Figure 4. Indeed, constant values of the phase fractions in the three-phase region of Figure 4 are guaranteed in a single point shown in region (α + β + γ) of Figure 4. Thus, the new definition is confirmed.

As shown above, all contradictions are resolved if the definition of the degree of freedom is extended as shown above. It is equally important to note that no new contradictions are created by this extension. As explained, there are the following three cases in Table 1: (i) cases with the phase fractions of all the single phases = 1: this value of 1 remains constant and valid in the whole region of *F* = 1, thus the above extension does not lead to any contradiction, (ii) cases with non-calculable phase fractions: as these phase fractions cannot be calculated, the above extension cannot lead to any contradiction, (iii) cases with calculable phase fractions being different from unity: the above extension was introduced to resolve the contradictions found for these cases without the extension.

## 11. Application of the General Phase Rule to Interpret Some Unusual Phase Diagrams

In addition to the classical case of $N_{NC-ISV}$ = 2 (*p, T*), further variable non-compositional state variables can also be selected, leading to some unusual phase diagrams or their sections. First, let us write the classical equation for the partial molar Gibbs energy of component i in phase Φ:(10a)$$G_{m,i\left(\Phi \right)}=U_{m,i\left(\Phi \right)}+p\cdot V_{m,i\left(\Phi \right)}-T\cdot S_{m,i\left(\Phi \right)}+G_{m,i(\Phi )}^{mag-in}$$
where $U_{m,i\left(\Phi \right)}$ (J/mol) is the partial molar inner energy of component i in phase Φ, $V_{m,i\left(\Phi \right)}$ (m^3^/mol) is the partial molar volume of component i in phase Φ (see models [31,32,33,34,35,36,37,38,39,40,41]), $S_{m,i\left(\Phi \right)}$ (J/molK) is the partial molar entropy of component i in phase Φ [31] and $G_{m,i(\Phi )}^{mag-in}$ (J/mol) is the partial molar Gibbs energy due to the inner magnetic moment of the atoms in a magnetic phase, without the effect of the external magnetic field [31,42,43,44,45,46,47,48]. Note: the inner magnetic term has a negative sign at low temperatures, i.e., it stabilizes the magnetic phase at low temperatures, but this stabilization effect gradually disappears with increasing temperature. This effect is not considered here, as it is not connected with any non-compositional state parameter. Moreover, the ferromagnetic-paramagnetic phase transition is not a first-order phase transition, meaning the phase rule discussed here is not valid for these two phases [29].

One can see that the two non-compositional state variables (*p* and *T*) used so far are coupled with some partial molar properties of the phase: pressure *p* (Pa) is coupled with $V_{m,i\left(\Phi \right)}$, while temperature *T* (K) is coupled with ${-S}_{m,i\left(\Phi \right)}$. Thus, Equation (10a) can be generally written as follows:(10b)$$G_{m,i\left(\Phi \right)}=U_{m,i\left(\Phi \right)}+G_{m,i(\Phi )}^{mag-in}+\sum_{j}G_{m,i(\Phi )}^{j}$$
where $G_{m,i(\Phi )}^{j}$ (J/mol) are different j terms of $G_{m,i\left(\Phi \right)}$ generally written as follows:(10c)$$G_{m,i(\Phi )}^{j}={SV}_{j}\cdot Y_{m,i(\Phi )}^{j}$$
where ${SV}_{j}$ is the state variable for the j-th term of $G_{m,i\left(\Phi \right)}$ and $Y_{m,i(\Phi )}^{j}$ is the partial molar property of component i of phase Φ in conjunction with the j-th state variable. The units of these quantities are different, but the complex units of all ${SV}_{j}\cdot Y_{m,i\left(\Phi \right)}^{j}\;$terms are J/mol. The following two j-terms are already written in Equation (10a):(10d)$$\mathrm{for}\;j=\mathrm{press}\;(\mathrm{the}\;\mathrm{pressure}\;\mathrm{term}):\;G_{m,i(\Phi )}^{press}=p\cdot V_{m,i\left(\Phi \right)}$$
(10e)$$\mathrm{for}\;j=\mathrm{temp}\;(\mathrm{the}\;\mathrm{temperature}\;\mathrm{term}):\;G_{m,i(\Phi )}^{temp}=T\cdot \left(-S_{m,i\left(\Phi \right)}\right)$$

Now, let us find the order of magnitude of a significant $G_{m,i(\Phi )}^{j}$-term. For metallic systems, the molar melting entropy has an order of magnitude of 10 J/Kmol-atom. Multiplying this by the minimum significant (measurable) melting temperature difference of 1 K, the value of $G_{m,i\left(\Phi \right),sign}^{j}$ = 10 J/mol is obtained as a minimum significant order of magnitude of the absolute value of the partial molar Gibbs energy from Equation (10e). The molar volume change upon melting is about 10^−6^ m^3^/mol. Substituting this latter value and $G_{m,i\left(\Phi \right),sign}^{j}$ = 10 J/mol into Equation (10d), p_sign_ = 10^7^ Pa = 100 bar. That is why phase equilibria between condensed phases are practically pressure-independent below 100 bar. As a result, considerable changes in phases equilibria due to high pressures start only above 100 bar = 0.01 GPa [31,32,33,49,50]. In the following sub-chapters, equations for further five terms of $G_{m,i(\Phi )}^{j}$ will be analyzed (see the summary Table 2).

### 11.1. The Electrochemical Term of the Partial Molar Gibbs Energy

The electrochemical term for the partial molar Gibbs energy is written as follows [1,51,52,53,54,55,56,57]:(10f)$$G_{m,i(\Phi )}^{el-chem}=-∆E\cdot F\cdot z_{i}$$
where ∆E (V) is the thermodynamic term of the electrochemical potential difference between the cathode and the anode (or the cathode and a reference electrode) through the electrolyte in the electrochemical system, F = 96,485 C/mol-electron is the Faraday constant (=the absolute charge of 1 mole of electrons = the absolute charge of 1 mole of protons = the absolute charge of 1 mole of single charged anions or cations), $z_{i}$ (dimensionless = mole electron per mole atom) is the number of electrons transferred through the system per number of electrodeposited atoms i. For this case ${SV}_{el-chem}=∆E$, while the molar property of the system is the product of $F\cdot z_{i}$ (C/mol-atom), substituting $G_{m,i\left(\Phi \right),sign}^{j}$ = 10 J/mol and the characteristic value of $z_{i}$ = 1 into Equation (10f), $∆E_{sign}$ = 10^−4^ V = 0.1 mV is obtained. Thus, phase equilibria during electrodeposition are very sensitive to the electrochemical potential difference.

The first electrochemical phase diagrams were the Pourbaix diagrams [58,59,60,61,62,63,64]. These diagrams have two axes with two independent *SV*s: the electrochemical potential discussed in this sub-chapter and the pH, the latter representing the concentration of H^+^ ions in aqueous solutions. The other potential *SV*s have fixed values: *p* = const, *T* = const, the concentration of metallic ions in the electrolyte = const. In accordance with Equation (5c), P_max_ = 1 + 2 − 0 = 3 is also valid for one-component Pourbaix diagrams. The Pourbaix diagrams are suitable only to aqueous solutions with dissolved hydrogen ions and to potentiostatic electrodeposition.

Another method of electrodeposition is the galvanostatic process in which current density is controlled. However, current density is not a thermodynamic state parameter, it is rather a kinetic parameter (=rate of charge transfer = rate of electron transfer ~ rate of electrochemical deposition). The phase diagrams, i.e., the equilibrium states of the metallic alloy during galvanostatic electrodeposition, are valid at negligible current densities. That is why the state variable for the previous version of electrochemical diagrams of galvanostatic processes is the electrochemical potential discussed here and the composition of the electrodeposited alloy [65,66,67,68,69]. However, in the new type of galvanostatic electrochemical diagrams, the electrochemical potential is a hidden state parameter, while the real SVs are the composition of the electrolyte and the composition of the alloy electrodeposited from the given electrolyte at T = const and p = const [70]. In this case, in accordance with Equation (5c), P_max_ = 2 + 1 − 0 = 3 is also valid for a binary electrodeposited alloy. The following rules were created to construct the new type of electrochemical diagrams [70]:(i).From among all different possible cathodic phases, the one will be electrodeposited that exhibits the least negative cathodic deposition potential (see the negative sign of Equation (10f));(ii).For binary A-B alloys, two or three (but never four or more—see the phase rule above) phases can be co-deposited on the cathode if their equilibrium cathodic potentials are equal and are the least negative ones among all possible cathodic phases or phase combination;(iii).The equilibrium compositions of solid and liquid A-B alloy solutions that form on the cathode can be found from the condition of equality of the partial equilibrium cathodic potentials of the two A-B components in the given solution phase.

Note that the cathodic deposition potentials mentioned in the above rules are measured relative to the anodic potential, while the latter is taken as a constant and independent of the nature and composition of the cathodic product. The above rules make it clear that phase equilibria under electrodeposition are mostly determined by the electrochemical potential, even if the latter does not appear along the axes of the electrochemical diagrams [70].

### 11.2. The Nano-Size Term of the Partial Molar Gibbs Energy

The nano-size term for the partial molar Gibbs energy is written as [1,71,72,73,74,75,76] (see also [77,78,79,80,81,82,83,84,85,86,87,88,89,90,91,92]):(10g)$$G_{m,i(\Phi )}^{nano}=A_{sp,\Phi }\cdot V_{m,i\left(\Phi \right)}\cdot \sigma_{\Phi /X}$$
where $A_{sp,\Phi }$ (m^2^/m^3^ = 1/m) is the specific surface area of the nano-phase Φ defined as the ratio of its surface area to its volume, $\sigma_{\Phi /X}$ (J/m^2^) is the interfacial energy between phase Φ and its surrounding phase X. Models and databanks on interfacial energies can be found in [93,94,95,96,97,98,99,100,101,102,103,104,105,106,107,108,109,110,111,112,113,114] and references thereof. Let us mention that $\sigma_{\Phi /X}$ is a function of curvature at its very high values [115,116,117,118,119,120], the latter being proportional to the specific surface area for curved phases. Thus, Equation (10g) could be written in a differential form for higher precision. Note: in Equation (10g), the interfacial energy of phase Φ is applied instead of the partial interfacial energy of the component $\sigma_{i,\Phi /X}$, due to the validity of the Butler equation ($\sigma_{i,\Phi /X}=\sigma_{\Phi /X}$) [121,122,123,124,125,126,127,128,129,130,131,132,133,134,135,136,137,138,139,140,141,142,143]. Nevertheless, the partial molar property of the component in the given phase for this case is $V_{m,i\left(\Phi \right)}\cdot \sigma_{\Phi /X}$, while the state variable for this case is $A_{sp,\Phi }$.

Let us also note that Equation (10g) is different from the classical Kelvin equation in which the curvature of the phase is used instead of its specific surface area. How to derive Equation (10g) and the many reasons why the Kelvin equation is wrong were discussed in details in the previous papers of the author [71,72,73,75,76] (see also [144,145]). Shortly, Equation (10g) follows from one of the equations of Gibbs [1]: dG=σ·dA. After integrating this equation, dividing the result by the amount of matter in the phase and remembering that the latter is the ratio of the volume of the phase to its molar volume, Equation (10g) follows.

Substituting $G_{m,i\left(\Phi \right),sign}^{j}$ = 10 J/mol and the characteristic values of $\sigma_{\Phi /X}$ = 0.01 … 0.1 J/m^2^ and $V_{m,i\left(\Phi \right)}$ = 10^−5^ m^3^/mol into Equation (10g), the possible significant interval of the specific surface area is as follows: $A_{sp,sign}$ = 10^7^ … 10^8^ 1/m. Substituting this value into the equation for the specific surface area of a sphere $A_{sp}=6/d$, the significant diameter of the sphere is obtained in the following possible interval: d_sign_ = 6 10^−8^ … 6 10^−7^ m = 60 … 600 nm. Thus, at and below the magnitude of 100 nm, the size of nano-phases has a significant effect on phase equilibria. The positive sign in Equation (10g) means that nano-phases are usually less stable than macro-phases, as interfacial energies have generally positive values. However, under special conditions, nano-phases (but not macro-phases) might have negative liquid/liquid interfacial energies [146,147,148,149,150] leading to spontaneous emulsification [151,152,153,154,155,156,157,158,159,160]. Note that spontaneous emulsification happens not only in colloid chemistry but also in steel metallurgy [156,160]. Negative interfacial energies can take place also at grain boundaries of nanoparticles due to enhanced segregation [161,162,163,164,165], allowing the synthesis of nano-grained alloys stabilized against both coarsening and precipitation [165,166,167,168,169].

There are many phase diagrams published for nano-materials [144,170,171,172,173,174,175,176,177,178,179,180]. However, the majority of those phase diagrams look exactly the same as the macro-phase diagrams, only the equilibrium lines are shifted due to the nano-effect. In other words, in the majority of those papers, the extended phase rule due to the new SV is neglected/forgotten. The effect of the extended phase rule for nano-materials has been discussed in the papers of the present author for more than a decade [181,182,183] (see also [184,185,186,187,188]). It was shown on the example of nano-Tl that four phases can be in equilibrium for a single component at special values of pressure, temperature and particle size [181]. Indeed, if C = 1 and $N_{NC-ISV}$ = 3 (p, T, A_sp_) are substituted into Equation (5c), $P_{max}=1+3-0$ = 4 is obtained. Later, it was also shown in agreement with experiments that nano-equilibria in unary systems include separated solidus and liquidus lines due to an additional SP [182,183], despite the fact that in macroscopic unary phase diagrams, the solidus and liquidus lines are merged into a single melting/solidification line. For the same reason, the usual tie-line construction becomes invalid for binary nano-systems.

Let us note that Equation (10g) is written for the simplest case when a nanophase Φ is usually surrounded by a single vapor phase. However, in reality, nano-phases are found in more complex situations. The corresponding equations are summarized in Table 3 [76].

The major message of Table 3 is that nano-phases do not have their own intrinsic properties including their intrinsic equilibrium state; their properties and equilibrium state rather depend on their surrounding phases. Let us note that the hidden potential in equations of Table 3 have not been realized yet, i.e., no papers with corresponding phase equilibria have been published. The only exception is the earlier paper by Joonho Lee et al. [174], who showed how phase equilibrium in a liquid sessile droplet depends on the substrate that holds the drop (see also [189]). However, in these papers, the extended phase rule for nano-materials was not discussed.

Let us note that the phase rule is similarly altered in colloid systems with nano-sized precipitates, when the size (specific surface area) of the nano-particles act as new non-compositional independent state variable [190,191].

### 11.3. The External Magnetic Term for the Partial Molar Gibbs Energy

The external magnetic term for the partial molar Gibbs energy is written as [192,193,194]:(10h)$$G_{m,i(\Phi )}^{mag-ex}=-B\cdot V_{m,i(\Phi )}\cdot M_{i(\Phi )}$$
where *B* (T) is the applied external magnetic field, $M_{i(\Phi )}$ (J/Tm^3^ = A/m) is the partial magnetization of component i in phase Φ. The *SV_j_* in this case is the applied external magnetic field *B* (T), while the partial molar property of the component in the given phase is $V_{m,i(\Phi )}\cdot M_{i(\Phi )}$ (J/Tmol). Let us mention that $M_{i(\Phi )}$ is generally a function of *B* and also temperature, so Equation (10h) could be written in a differential form for higher precision. The negative sign in Equation (10h) means that phases of higher magnetization at given external magnetic field have more negative molar Gibbs energy values and thus are more stable compared to phases with lower magnetization values. Similarly to the inner magnetic term, this external magnetic term also gradually reduces in its absolute value with increasing temperature. More details are given in [4,29,192,193,194].

Substituting $G_{m,i\left(\Phi \right),sign}^{j}$ = 10 J/mol, the characteristic values of $M_{i(\Phi )}$ = 10^5^ … 10^6^ J/Tm^3^ and $V_{m,i(\Phi )}$ = 10^−5^ m^3^/mol into Equation (10h), the possible orders of magnitude of the significant external magnetic field are as follows: B = 1 … 10 T. Thus, the external magnetic field has an influence on phase equilibria starting from about 1 T and above.

The actual effect of the external magnetic field on phase equilibria has been found in the number of papers both experimentally [195,196,197,198,199] and theoretically [194,200,201,202,203,204,205]. Substituting C = 2 for a binary system and $N_{NC-ISV}$ = 2 for independent state variables *T* and *B* at *p* = const into Equation (5c), P_max_ = 2 + 2 − 0 = 4. Instead of complicated 3D diagrams (with *x_B_, T* and *B* along its axes), phase equilibria are usually shown along their 2D sections with variables *x_B_* and *T* at *p* = const and *B* = const. At these sections, only triple points (eutectic, peritectic, etc.) with P_max_ = 3 can be found. However, a unique combination of special values of *x_B_, T* and *B* (at *p* = const) can be found with *P_max_* = 4. An example can be found in the paper by Zeng et al. [194] for the estimated independent *SV* values of a four-phase point in the Bi-Mn system at *x_Mn_* = 0.51, *T* = 719.6 K and *B* = 48.56 T (at *p* = const = 1 bar) in agreement with the general phase rule derived in this paper.

### 11.4. The Elastic Term of the Partial Molar Gibbs Energy

The elastic term for the partial molar Gibbs energy is written as [206,207,208,209,210,211,212,213]:(10i)$$G_{m,i(\Phi )}^{elast}=V_{m,i(\Phi )}\cdot \omega_{i(\Phi )}\cdot \varepsilon^{2}$$
where $\omega_{i(\Phi )}$ (Pa = J/m^3^) is the complex elastic constant of component i in phase Φ, ε (dimensionless, m/m) is the relative linear strain (extension = positive strain, contraction = negative strain) defined as $\left(L-L_{o}\right)/L_{o}$ (where L (m) is the actual length with strain and $L_{o}$ (m) is the initial length with no strain). Note: the relative volume strain is about three times the relative linear strain due to three dimensions of the volume relative to the one dimension of the length. The formal derivation of Equation (10i) is simple: the elastic energy density is the strain multiplied by stress, while the stress is proportional to the strain according to Hook’s law, and so the elastic energy density is an elastic constant times the square of the strain, while the molar elastic energy of Equation (10i) is obtained by multiplying the latter by the molar volume. Due to the square of strain in Equation (10i), the elastic Gibbs energy always has positive values, i.e., strain de-stabilizes phases. The order of magnitude of the complex elastic constant is the same with the bulk modulus of the phase. However, the exact equation for the complex elastic constant of Equation (10i) is really complex: it depends on the shape and relative orientation of the strain and the crystal planes; moreover, for two-phase systems, it also depends on the relative orientation of the two phases and their elastic constants [206,207,208,209,210,211,212].

The *SV_j_* for this case is the strain ε (dimensionless), while the partial molar property of the component in the given phase is $V_{m,i(\Phi )}\cdot \omega_{i(\Phi )}$ (J/mol). Substituting $G_{m,i\left(\Phi \right),sign}^{j}$ = 10 J/mol, the characteristic values of $V_{m,i(\Phi )}$ = 10^−5^ m^3^/mol and $\omega_{i(\Phi )}$ = 10…1000 GPa into Equation (10i), $\varepsilon_{sign}$ = 0.001 … 0.01 is obtained. It means that the elastic energy has a significant effect on phase equilibria starting from 0.1% of strain, and it becomes really significant above 1% of strain.

The elastic strain energy written by Equation (10i) plays a major role in the nucleation and stabilization of nano-precipitates from over-saturated solid solutions. Nucleation of a nano-phase is easier and its stabilization is enhanced when a precipitate forms a coherent interface with the matrix. However, for ideal coherency, perfect matching of the crystal lattices of the two phases is requested. Although perfect matching is practically impossible, close matching happens and in these cases induced coherency is ensured by the spontaneous development of strain within and around the nano-precipitate. The elastic energy developed in this way inside and around the nano-phase influences the phase equilibrium between the matrix and the precipitate, called “coherent equilibrium” [209,213,214,215,216,217,218,219,220,221]. For simplicity, the interfacial energy is usually neglected in those calculations. However, there is always some interplay between the nano-size effect and the elastic strain. As a result, precipitates usually nucleate as spheres (to minimize their specific interface area), but if the matrix is cubic, then the precipitate upon its coarsening will be transformed into a cubic phase [222] as it has the lowest elastic constant in a cubic matrix if properly oriented [209]. This is because the surface effect is proportional to r^2^, while the elastic effect is proportional to r^3^ (where r is the characteristic size of the nano-phase) [213,222]. This theoretical finding was also reproduced by computer simulations [223,224,225,226]. The effects of the nano-size term and the elastic term are so closely related that the “effective interfacial energy of strained and coherent nanoparticles” could be recently derived [222], i.e., the interfacial energy of the strain-free coherent nano-particle [112] was increased due to the strain developed by the misfit between the two phases [222]. These two effects cannot be distinguished when the effective interfacial energy is obtained through fitting a coarsening model to experimental coarsening data [227].

It is claimed in several papers that the original phase rule of Gibbs written by Equations (1a) and (1b) is not valid under elastic strain [14,209,210,214,215] and so the usual tie-line construction cannot be applied. This is because a new state parameter (ε) should be taken into account, which modifies the phase rule as written by Equation (5c), also making the tie-line construction invalid (see a similar comment above, related to the nano-size effect).

### 11.5. The Gravitational Term of the Partial Molar Gibbs Energy

The gravitational term for the partial molar Gibbs energy follows from the potential energy of Newton written for the partial mass of a component in a phase divided by the amount of that component in that phase [1]:(10j)$$G_{m,i}^{grav}=h\cdot g\cdot M_{m,i}$$
where *h* (m) is the relative elevation above some standard height level in a gravitational field, *g* (m/s^2^) is the local acceleration due to gravity and $M_{m,i}$ (kg/mol) is the molar mass of component i. Note that phase Φ is not mentioned here, as molar masses of components are practically independent of the phase. This is because the interaction energy between the atoms within molecules and phases is so weak that it does not lead to any measurable variations in the values of atomic masses according to the modified Einstein equation ($∆M=∆G/c^{2}$, where c is the velocity of light).

The *SV_j_* in Equation (10j) is h·g, while the partial molar property of the component is $M_{m,i}$. Substituting $G_{m,i\left(\Phi \right),sign}^{j}$ = 10 J/mol and the interval of characteristic values $M_{m,i}$ = 0.001 … 0.1 kg/mol into Equation (10j), the possible interval of the significant value follows as ${\left(h\cdot g\right)}_{sign}$= 100 … 10,000 m^2^/s^2^. Taking into account g = 9.81 m/s^2^ on the surface of the Earth, one can conclude that gravity will have any effect on phase equilibria of condensed phases only in a vessel significantly higher than 10 m (such industrial vessels are quite rare). Note: as the surface acceleration due to gravity is smaller on both the Moon and Mars compared to the Earth, mankind is not going to face the problem of gravity influencing phase equilibria very soon.

The effect of gravity due to Equation (10j) is somewhat important only for the equilibrium distribution of different components of different molar masses in high columns of fluids at the scale of km-s, such as in the atmosphere of the Earth, or in reservoirs’ fluids [228,229,230]. As follows from Equation (10j), components with large molar masses have more positive partial molar Gibbs energies at high elevations compared to components of lower molar masses, and so “heavy components” are concentrated at low altitudes, while “light components” are concentrated at high altitudes in the atmosphere of the Earth (note: this rule was already known by some of the ancient Greek philosophers).

### 11.6. Summary on the Phase Rule Applied to Unusual Phase Diagrams

Equations (10a)–(10j) can be summarized as follows:(10k)$$G_{m,i\left(\Phi \right)}=U_{m,i\left(\Phi \right)}-T\cdot S_{m,i\left(\Phi \right)}+G_{m,i\left(\Phi \right)}^{mag-in}-∆E\cdot F\cdot z_{i}+h\cdot g\cdot M_{m,i}+\;+V_{m,i\left(\Phi \right)}\cdot \left(p+A_{sp}\cdot \sigma_{\Phi /X}+B\cdot M_{i\left(\Phi \right)}+\varepsilon^{2}\cdot \omega_{i(\Phi )}\right)$$

It should be noted that Equation (10k) is the simplest form of the partial molar Gibbs energy with seven independent *SV*s and nine additive terms, as all terms are written as independent, additive terms. However, these terms are probably interrelated. Thus, it is a real challenge to create the model equation for $G_{m,i\left(\Phi \right)}=f(x_{i\left(\Phi \right)},\;T,\;p,∆E,\;h,\;A_{sp},\;B,\;\varepsilon )\;$as the function of seven non-compositional state parameters. This equation is needed as a solid theoretical basis to exploit the potential hidden in all external fields to create more advanced materials.

A deeper exploitation of old and new non-compositional state parameters is very important in the development of advanced materials to compensate for the gradual decrease in available compositional state parameters. This decrease takes place as both medical sciences and the complexity of international affairs gradually develop in time, leading to ever-shorter lists of non-toxic elements and to ever-longer lists of critical elements from the point of view of political economics.

Application of machine learning in constructing phase diagrams is an important new sub-field. However, for higher efficiency, those algorithms should be coupled with the phase rule [231]. The future machine learning algorithms should be coupled with the extended phase rule as shown in this paper.

## 12. Conclusions

The component rule is introduced in this paper as follows: “The maximum number of independent components to be selected for phase equilibria calculations from a list of species of interest (elements and compounds) equals the total number of chemical elements in all those species”. If at least one of the components is selected as a compound instead of an element, care should be taken to ensure that no chemical reaction exists between the selected components and also that their composition range includes the full compositions range of all the selected species of interest. If the latter two conditions are obeyed, a smaller than the maximum number of components can also be selected for convenience (although in this case usually some information is lost).It is explained that the two classical forms of the phase rule due to Gibbs ($P_{max}=C+2$ and F=C+2−P) are valid only if there are two non-compositional, independent state variables (such as pressure and temperature) and if no compositional constraints are enforced. The strange practice in the literature is highlighted that almost each time before these equations are applied, they are modified for the given situation. This paper offers equations that can be applied without a need to modify them each time before they are used.The independent state variables are divided here into two groups: into the compositional ones and into non-compositional ones (their number is denoted as $N_{NC-ISV}$). The general phase rule of Gibbs is derived for this case as $P_{max}=C+N_{NC-ISV}-Z_{NC}$ and $F=C+N_{NC-ISV}-Z_{NC}-P$, i.e., number 2 in the original phase rule is replaced by $N_{NC-ISV}-Z_{NC}$, where $Z_{NC}$ is the number of constraints enforced between the non-compositional state variables. This generalized phase rule can be used without a need to change it each time before it is applied. Examples are cited from the literature to show the validity of these general equations when different independent non-compositional state variables are applied. It is shown that the same equation is valid for the quasi-binary sections of the multi-component phase diagrams when constraints are applied between the average mole fractions of components in the system.It is also shown that the same equation ($P_{max}=C+N_{NC-ISV}-Z_{NC}$) is also valid when constraints are applied between the equilibrium mole fractions of equilibrium phases in special points of phase diagrams, such as in the azeotrope points or in congruent melting points. It is also shown that in these special points, the degree of freedom is decreased further by the number of such constraints.The definition of the degree of freedom is extended as follows: “the degree of freedom (*F*) is the number of independent state variables whose values can be freely varied in a finite interval without the appearance or disappearance of a phase and without changing the selected equilibrium values of the phase fractions of equilibrium phases” (the latter is the extension).It is shown that in all 2D phase diagrams or in all 2D phase diagram sections, *P_max_* = 3 is always valid, except in the case of quasi-binary vertical sections of ternary or multi-component phase diagrams with constraints between the average mole fractions of components in the system.In addition to the two classical non-compositional state variables (pressure and temperature), equations are provided for partial molar Gibbs energies as the function of the further five non-compositional state variables: the electrode potential difference, the specific surface area of a nano-phase, the external magnetic field, the elastic strain, and the relative elevation in the gravitational field. Examples are given from the literature to show that our general phase rule is valid for all these cases.

## 13. Nomenclature

In this section, terms with their definitions, abbreviations with their explanations and symbols of physical quantities with their units and definitions are collected that are essential to follow this paper.


**Terms and Their Definitions**


Characteristics of the state: number and identity of phases, their compositions and phase fractions; although their equilibrium values are determined by the state parameters via the laws of Nature, the engineer only has technical ways to set the values of state parameters.

Components: selected elements and/or compounds from the list of species of interest; they determine phase equilibria.

Components, their maximum number: the maximum number of components that can be selected from the given list of species of interest defined by the component rule.

Component rule: the maximum number of components to be selected for phase equilibria calculations from a list of species of interest (elements and compounds) equals the total number of elements contained in all those species of interest.

Constraints, compositional: mathematical relations enforced by the engineer between average mole fractions of components.

Constraints, non-compositional: mathematical relations enforced by the engineer between non-compositional state variables.

Degree of freedom: the number of independent state variables whose values can be freely varied in a finite interval without the appearance or disappearance of a phase and without changing the selected equilibrium values of the phase fractions of equilibrium phases.

Molar Gibbs energy, partial of a component: a property of a component in a phase; in equilibrium, each component has equal values in all phases and in a whole system (also called chemical potential).

Molar Gibbs energy, integral of a phase: the weighted (by molar ratios) average partial molar Gibbs energies of all components in a given phase: a property of a phase.

Molar Gibbs energy, average of the system: the weighted (by phase fractions) average of integral molar Gibbs energies of all phases in a system: its minimum corresponds to the equilibrium in the system.

Molar Gibbs energy, significant: the minimum (in absolute values) value of the molar Gibbs energy that has a significant influence on phase equilibria.

Molar ratio of components, average: the amount of matter of a given component in a system divided by the total amount of matter in the same system (a state parameter).

Molar ratio of components, in a phase: the amount of matter of a given component in a phase divided by the total amount of matter in the same phase (not a state parameter, rather one of the characteristics of the state).

Phase: a 3D piece of homogeneous material that has a given structure and composition that makes it different from other phases.

Phase combinations: all the possible phases or combinations of two or more phases that can possibly coexist in equilibrium under set values of state parameters; to make the list of phase combinations is an essential part to calculate phase equilibria.

Phase diagram: a diagram that has independent state variables plotted along its axes and shows regions of equilibrium phases or phase combinations.

Phase diagram, its section: when phase equilibria are studied along more than two independent state variables, it is convenient to show different sections of the phase diagram along which one or more state variables have fixed values, or fixed ratios of values.

Phase fraction: amount of matter of a given phase divided by the total amount of matter in the system that contains the given phase (not a state parameter, rather one of the characteristics of the state).

Phase rule: a simple rule connecting the number of components, the number of non-compositional state variables, the number of constraints between the latter and the equilibrium number of phases with the maximum number of coexisting phases and with the degree of freedom.

Phases, their maximum number: this is the maximum number of different, coexisting phases that can be found experimentally in a system.

Species (of interest): elements or chemical compounds of interest for the engineer; components are selected from the list of species of interest, but not necessarily all species of interest are selected as components.

State parameters: physical or chemical quantities that have influence on the equilibrium state and whose values are selected by the engineer and can be enforced on the system by the engineer in a technical way.

State variables: state parameters whose values are considered variables (i.e., whose values are not fixed); independent state variables are those state variables that vary independently of each other.

State variables, compositional: the average mole fractions of components in a system.

State variables, non-compositional: state variables except the compositional ones: usually pressure and temperature, but see Section 11 for more.

State variables, of free values: whose values can be freely changed by an engineer within a finite interval without changing the number and identity of equilibrium phases (= the degree of freedom).

State variables, of fixed values: whose values should be kept constant to make sure that the number and identity of equilibrium phases is not changed (their constant values follow from the laws of Nature).

States: characterized by the number and identity of phases, their compositions and fractions; the laws of Nature drive the system from its initial state (as designed by the engineer at the beginning of the experiment) towards the equilibrium state that is approached asymptotically in time.

System: an imaginary complex material usually defined by the number and identity of its components; in reality, it is a 3D object, being single phase or the mixture of two or more phases that contain the dissolved components.


**Abbreviations and Their Explanations**


2D or 3D: 2-dimensional or 3-dimensional.

A or B or C: symbols for the component.

*g* = gas.

i = general symbol of a component.

*l* = liquid.

*v* = vapor.

Calphad = Calculation of phase diagrams.

ISV = independent state variable.

NC-ISV = non-compositional independent state variables.

SP = state parameter (or special point in Figure 3).

SV = state variable.

${SV}_{j}$ = state variable of the j-term of $G_{m,i\left(\Phi \right)}$.

${SV}_{j,sign}$ = the significant value of the state variable of the j-term of $G_{m,i\left(\Phi \right)}$.

$Y_{m,i(\Phi )}^{j}$: the partial molar property for the j-term of $G_{m,i\left(\Phi \right)}$.

α or β or γ: symbols for phases.

Φ: general symbol for a phase.


**Physical Quantities with Their Units and Definitions**



**
*Symbol*
**

**Unit**

**Definition**

*A*
m^2^surface area

$A_{sp,\Phi }$

1/mspecific surface area of a phase
*B*
Tapplied external magnetic field
*C*
--number of components in a system
*EQ*
--number of independent equations type (3a)–(4c)
*EQ_MB_*
--number of materials balance equations type (9a)
*F*
C/mol-ethe Faraday number (=96,485 C/mol-e)
*F*
--degree of freedom
*F**
--degree of freedom in special points of a phase diagram
*g*
m/s^2^acceleration due to gravity

$G_{m,i\left(\Phi \right)}$

J/molpartial molar Gibbs energy of a component in a phase

$G_{m,i(\Phi )}^{j}$

J/mol

$\mathrm{the}\;j-\mathrm{term}\;\mathrm{of}\;G_{m,i\left(\Phi \right)}$



$G_{m,i\left(\Phi \right),sign}^{j}$

J/mol

$\mathrm{the}\;\mathrm{significant}\;\mathrm{value}\;\mathrm{of}\;G_{m,i\left(\Phi \right)}^{j}$



$G_{m,i(\Phi )}^{mag-in}$

J/mol

$\mathrm{the}\;\mathrm{inner}\;\mathrm{magnetic}\;\mathrm{term}\;\mathrm{of}\;G_{m,i\left(\Phi \right)}$



$G_{m,i(\Phi )}^{press}$

J/mol

$\mathrm{the}\;\mathrm{pressure}\;\mathrm{term}\;\mathrm{of}\;G_{m,i\left(\Phi \right)}$



$G_{m,i(\Phi )}^{temp}$

J/mol

$\mathrm{the}\;\mathrm{temperature}\;\mathrm{term}\;\mathrm{of}\;G_{m,i\left(\Phi \right)}$



$G_{m,i(\Phi )}^{el-chem}$

J/mol

$\mathrm{the}\;\mathrm{electrochemical}\;\mathrm{term}\;\mathrm{of}\;G_{m,i\left(\Phi \right)}$



$G_{m,i(\Phi )}^{nano}$

J/mol

$\mathrm{the}\;\mathrm{nano}-\mathrm{size}\;\mathrm{term}\;\mathrm{of}\;G_{m,i\left(\Phi \right)}$



$G_{m,i(\Phi )}^{mag-ex}$

J/mol

$\mathrm{the}\;\mathrm{external}\;\mathrm{magnetic}\;\mathrm{term}\;\mathrm{of}\;G_{m,i\left(\Phi \right)}$



$G_{m,i(\Phi )}^{elast}$

J/mol

$\mathrm{the}\;\mathrm{elastic}\;\mathrm{term}\;\mathrm{of}\;G_{m,i\left(\Phi \right)}$



$G_{m,i(\Phi )}^{grav}$

J/mol

$\mathrm{the}\;\mathrm{gravitational}\;\mathrm{term}\;\mathrm{of}\;G_{m,i\left(\Phi \right)}$


*h*
mrelative elevation above some standard height
*L*
mthe actual length of a body with strain

$L_{o}$

mthe initial length of a body with no strain

$M_{m,i}$

kg/molthe molar mass of a component

$M_{i(\Phi )}$

J/Tm^3^the partial magnetization of a component in a phase
*N_ISV_*
--number of independent state variables
*N_ISV-FIX_*
--number of independent state variables of fixed values

$N_{ISV-FIX}^{*}$

--same as *N_ISV-FIX_* in special points of phase diagrams
*N_NC-ISV_*
--number of independent non-compositional state variables
*p*
Papressure (one of the state parameters)
*P*
--number of phases in a system
*P_max_*
--the maximum number of coexisting phases in a system

$P_{max-red}$

--$\mathrm{reduced}\;P_{\max}=\mathrm{the}\;\mathrm{same}\;\mathrm{as}\;P_{\max}\;\mathrm{at}\;Z_{C}$ = 0
*PAR*
--$\mathrm{number}\;\mathrm{of}\;\mathrm{ISVs}\;\mathrm{in}\;\;G_{m,i(\Phi )}=f(x_{i\left(\Phi \right),}T,\;p,\;\ldots )$ functions
*PAR**
--same as *PAR* in special points of phase diagrams
*r*
mcharacteristic size of a phase

$r_{c}$

mradius of a cylinder

$S_{m,i\left(\Phi \right)}$

J/molKpartial molar entropy of a component in a phase
*T*
Ktemperature (one of the state parameters)

$U_{m,i\left(\Phi \right)}$

J/molpartial molar inner energy of a component in a phase

$V_{m,i\left(\Phi \right)}$

m^3^/molpartial molar volume of a component in a phase
*x_i_*
--average mole fraction of component i in a system

$x_{i\left(\Phi \right)}$

--

equilibrium mole fraction of component i in phase Φ



$y_{\Phi }$

--phase fraction of phase Φ in a system
*z_i_*
--(mol-e/mol-i)the number of electrons transferred per number of electro-deposited atoms i
*Z_C_*
--number of compositional constraints
*Z_NC_*
--number of non-compositional constraints
*Z_P_*
--number of constraints at special points

∆E

Vthermodynamic term of the electrochemical potential 

ε

-- (m/m)relative linear strain

$\sigma_{\Phi /X}$

J/m^2^the interfacial energy between phases Φ and X

$\sigma_{i,\Phi /X}$

J/m^2^the partial interfacial energy between phases Φ and X

$\omega_{i(\Phi )}$

J/m^3^complex elastic constant of a component in a phase

Θ

degreethe contact angle of a liquid on a solid in gas

## Figures and Tables

**Figure 1 materials-17-06048-f001:**
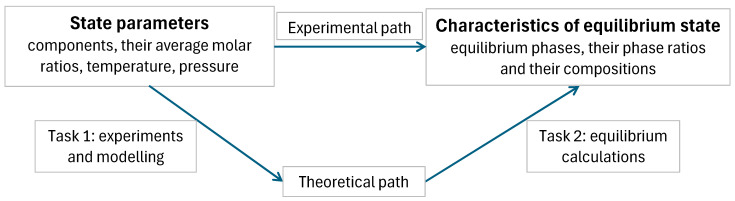
The schematic algorithm to find phase equilibria from the state parameters.

**Table 2 materials-17-06048-t002:** The seven non-compositional state variables and their conjugant partial molar quantities for different terms of the partial molar Gibbs energy of Equation (10c).

Term	${SV}_{j}$	Unit	SV_j,sign_	$Y_{m,i(\Phi )}^{j}$	Unit	Equation
pressure	p	Pa	10^7^	$V_{m,i\left(\Phi \right)}$	m^3^/mol	(10d)
temperature	T	K	1	$-S_{m,i\left(\Phi \right)}$	J/molK	(10e)
electrochemical	∆E	V	10^−4^	$-F\cdot z_{i}$	C/mol	(10f)
nano-size	$A_{sp,\Phi }$	1/m	10^7^	$V_{m,i\left(\Phi \right)}\cdot \sigma_{\Phi /X}$	Jm/mol	(10g)
external magnetic	B	T	1	$M_{i(\Phi )}\cdot V_{m,i(\Phi )}$	J/Tmol	(10h)
elastic	ε	m/m	10^−3^	$V_{m,i(\Phi )}\cdot \omega_{i(\Phi )}$	J/mol	(10i)
gravitational	g·h	m^2^/s^2^	100	$M_{m,i}$	kg/mol	(10j)

**Table 3 materials-17-06048-t003:** Equations similar to Equation (10g) for nano-phases in complex situations [76] *.

Nano-Phase	Situation	Equation
liquid drop	in its vapor	$G_{m,i(l)}^{nano}=A_{sp,l}\cdot V_{m,i\left(l\right)}\cdot \sigma_{l/g}$
liquid drop 1	in another liquid 2	$G_{m,i(l1/l2)}^{nano}=A_{sp,l1}\cdot V_{m,i\left(l1\right)}\cdot \sigma_{l1/l2}$
sessile drop	on a solid substrate	$G_{m,i(l/s)}^{nano}=G_{m,i(l)}^{nano}\cdot {\left(\frac{2-3\cdot cos\Theta +{cos}^{3}\Theta }{4}\right)}^{1/3}$
solid particle	in its vapor	$G_{m,i(s)}^{nano}=A_{sp,s}\cdot V_{m,i\left(s\right)}\cdot \sigma_{s/g}$
solid particle	at liquid/gas surface	$G_{m,i(s-l/g)}^{nano}=G_{m,i(s)}^{nano}\cdot \left[1-\frac{\sigma_{l/g}}{\sigma_{s/g}}\cdot {\left(\frac{1+cos\Theta }{2}\right)}^{2}\right]$
liquid	in cylindrical capillary	$G_{m,i(l/cap)}^{nano}=-\frac{2}{r_{c}}\cdot V_{m,i\left(l\right)}\cdot \sigma_{l/g}\cdot cos\Theta$

* Θ (degree) is the contact angle of a liquid on a solid in gas; r_c_ (m) is the radius of the cylindrical capillary.

## Data Availability

The original contributions presented in the study are included in the article, further inquiries can be directed to the corresponding author.

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
