# Peer review of "The Generalized Phase Rule, the Extended Definition of the Degree of Freedom, the Component Rule and the Seven Independent Non-Compositional State Variables: To the 150th Anniversary of the Phase Rule of Gibbs"

_materials, 2024, doi:10.3390/ma17246048_

Round 1
Reviewer 1 Report
Comments and Suggestions for Authors
Please see the attached file

Author Response
Reply to the reviewers of the paper:
The generalized phase rule, the extended definition of the degree of freedom and the component rule. To the 150th anniversary of the phase rule of Gibbs by George Kaptay, submitted to Materials
Dear Editor and Reviewers,
I thank all of you for your time and attention to my paper. Herewith I number and reply all your comments and the corresponding changes are included in version R1 of the paper by yellow color (and some of them are also copied here in yellow color). I hope that this revised paper can be published soon in your journal.
Sincerely, George Kaptay, sole author
Reviewer 1.
Comment 1.1. The topic of this work deserves attention and a work like this could be of interest and useful for a broad range of researchers in different disciplines and fields of application. However, the exposition is not clear in some parts of the manuscript and there is room for improvements of different type. At some points, I had the impression that, with his arbitrary reasonings and definitions, instead of clarifying the subject, the author creates more confusion, for example in section 9 with the apparent contradictions. The following list of suggestions and observations should be considered before publication.
Reply 1.1. Thanks a lot. It is a strong and detailed review, which helped me a lot to improve my paper further.
Comment 1.2. Sections 2 to 6 are developed and written in a very abstract way. More examples could help the readers to follow the ideas and equations.
Reply 1.2. Thank you for this. I got a permission from the editor to enlarge the paper, and so 3 examples are added to Section 2. A new sentence is added to section 6 and actually section 7 is an explanation of section 6.
Comment 1.3. I have a problem with the “component rule” defined and proposed in section 2. When applied to a non-reacting mixture of N different hydrocarbons, the number of elements is just 2 (C and H) while the number of independent components is N, which can be any number, e.g. 5.
Reply 1.3. When you mix 5 different hydrocarbons, to my opinion you have 5 species and not five components, as only 2 of them are components in thermodynamic sense, that determine the number of possible phases. How to select the 2 components from the 5 species? I think it is better to select one with the lowest and the second one with the highest H-content, so these two components cover the whole composition range. Yes, I also understand your condition “non-reacting”. However, the phase rule of Gibbs does not consider kinetic constraints, slowing down the reaction / dissolution between the 5 CHx species. This idea is explained in Example 3 of section 2.
Comment 1.4. A nomenclature section is required, specially in a work like this, with so many defined variables.
Reply 1.4. Thank you for the suggestion, it is included at the end of the paper as the editor kindly agreed to increase the volume of the paper.
Comment 1.5. Please explain more clearly section 6, so one can see its meaning and usefulness. In particular, it sounds unreasonable to assign NISV = 2 just because we are considering a 2D diagram.
Reply 1.5. In principle, on a 2D page we can show diagrams with 3 axes (x-y-z) or even more. However, if we show 3 axes, then in this 3D diagram shown on a 2D page some lines and surfaces usually overlap and so we cannot see clearly and read accurately the position of each line and each surface. Therefore, people prefer 2D diagrams on 2D pages, and that is why I limit NISV = 2, i.e. two axes. I also added a new sentence to section 6: “The three examples presented in the next section (see Fig-s 2-3-4 and table 1) provide three different phase diagrams, which look different, but are similar in a sense that for all of them = 2 and Pmax-red = 3.”
Comment 1.6. In section 7: “4 phases cannot coexist in ternary systems with fixed pressure and temperature values.” This is simply wrong. Four-phase equilibrium has one degree of freedom for ternary systems. So, given a system presenting a certain type of four-phase equilibrium, if we fix T, then we can observe 4 phases in equilibrium at the corresponding pressure. It is just like three-phase equilibrium in a binary system. To give an example, just in case, diagrams with calculated four phase lines for a ternary system can be found in The Journal of Supercritical Fluids, Vol. 130, 399-414. 2017. ( DOI: 10.1016/j.supflu.2017.06.005 ). That publication also shows 4-phase equilibrium at given values of T and P, as part of a prism of constant T, like in the following figure:
Reply 1.6. This seeming contradiction is resolved as (see new text in the paper): “However, one should note that it does not mean that 4 phases cannot be present in ternary phase diagrams if plotted as function of compositions and pressure (see [30] as an example). Nevertheless, at a fixed value of pressure and at a fixed value of temperature = 0 and so from Eq.(5c): . The seeming contradiction between the 4 phases at a given pressure (at fixed temperature) in [30] and between the calculated value of can be resolved as: in real life there is a negligible probability to find the exact special pressure value (at fixed temperature) or the exact special temperature value (at fixed pressure) at which 4 phases indeed coexist in equilibrium in a ternary system. Let me note that this argument is already given by Gibbs [1].
Comment 1.7. What the author calls phase ratios are actually phase fractions, and that is how they are named in the literature, at least in Chemical Engineering Thermodynamics and related areas. See for example the book “Thermodynamic Models: Fundamentals and Computational Aspects” by Michelsen and Mollerup (Tie-Line Publications, 2007) and all the classic works by Michelsen.
Reply 1.7. Thanks for this. The quantity phase ratio is now renamed to phase fraction.
Comment 1.8. The work could be more interesting and useful if variables like overall molar volume, enthalpy or entropy were considered. Their specification can make the phase fractions calculable in the cases of Table 1 where they appear as not calculable.
Reply 1.8. Yes, you are right, phase fractions can indeed be calculated if overall molar volume, enthalpy or entropy are given. However, these quantities are not independent state variables and that is why they are not visible in usual phase diagrams. That is why I do not use them in this paper.
Comment 1.9. Considering the comment at the end of section 9… The proper variables of the system, the ones involved in the phase equilibrium equations, are the mole fractions in each equilibrium phase and not the global or average mole fractions. That is why the identified contradictions are just apparent and not true contradictions.
Reply 1.9. Here I do not agree, at least partly. Yes, it is true that the equilibrium mole fractions appear in equations for phase equilibria calculations. As a result, these values can be calculated from those equations, so they are some of the “parameters characterizing the equilibrium state”. But they are not independent state variables of the system; those are the independent (average) mole fractions of components in the system. We, engineers have technical ways to define only the state variables, including the average mole fractions of components in the system. Then we wait some time for the laws of Nature to drive the system towards equilibrium, then open the system and determine the number of phases + their identities + their compositions + their phase fractions. All these latter 4 are the characteristics of the equilibrium state of the system. Example: binary system A-B, let us fix as independent state variables xB = 0.5, T = 500 K, p = 2 bar. Suppose this leads to a 2-phase equilibrium, but we do not know this in advance. We open the system and find that we have two phases. Then we determine which phases they are (alpha and beta). Then we measure the phase fraction of phase beta (y-beta) and then we measure the mole fractions of component B in two phases: xB(alpha) and xB(beta). Of-course, the latter 3 are connected via the material balance equation. But the experimentally found results (number of phases, their identity, their compositions and their phase fractions) are characteristics of the equilibrium state, and they are not the independent state variables. Now, one page is added to Introduction to make this point clear (see Fig.1). A text is added: “Before doing so let me mention that in some texts the contradiction found for the above case of 2-component, 2-phase regions is wrongly “resolved” by claiming that the degree of freedom is 1, as at a freely selected temperature the equilibrium compositions of the equilibrium phases are fixed at the two ends of the tie-line and so the second degree of freedom has a fixed value (see [12], as an example). This reasoning is incorrect, because the degree of freedom should be selected from the list of independent state variables, but the equilibrium compositions of equilibrium phases are not in the list of state variables (note: the average mole fractions of the system are state variables – see Fig.1).“
Comment 1.10. Maybe the main contribution of this work could be in section 11, when cases with other noncompositional state variables are considered. Those situations may really justify and extension or different treatment of the phase rule. Therefore, I would suggest putting more focus and communication effort there, rather than on tangling things up for typical and simpler systems that can be understood easier (e.g. sections 9 and 10).
Reply 1.10. Thank you for this, now section 11 is enlarged, also thanks to the permission of the editor to enlarge the volume of the paper. The absolutely new text is written, but it is too long, so it is not copied here for brevity.
Comment 1.11. Another important application of the gravitational field effect can be on reservoirs fluids, as treated for example in the following books: a. Pedersen, K. S.; Christensen, P. L. Phase Behavior of Petroleum Reservoir Fluids; CRC Press (Taylor & Francis Group, 2006); DOI: 10.1201/9781420018257., b. Firoozabadi, A. Thermodynamics and Applications in Hydrocarbon Energy Production (McGraw-Hill, 2016) c. Esposito, R., Scilipoti, J., Alijo, P., Tavares, F. Compositional Grading in Oil and Gas Reservoirs (Elsevier, 2017)
Reply 1.11. Thanks for this, this comment is added.

Reviewer 2 Report
Comments and Suggestions for Authors
Review: materials-3285313
This article revisits the phase rule of Gibbs from the 1870s. Compelling arguments are presented with regard to the selection of independent components, independent state variables, and degrees of freedom. In addition, the author presents an interesting take on the relationship between the number of coexisting phases and the dimensions of the phase diagram of interest. In my opinion, this paper is rather compelling.
I have a couple of general comments and questions related to this work:
1. Can the author perhaps apply their approach to the system mentioned here: https://doi.org/10.1103/PhysRevLett.125.127803 as further evidence of its robustness?
2. Can the author comment on the implementation of their propositions in machine learning systems [e.g. https://doi.org/10.1016/j.scriptamat.2021.114335]. I suppose implementation would be straightforward; are there any plans to undertake such work in future?
Here are few minor comments or questions related to the manuscript:
3. Page 6, lines 264 & footnote: I’m not sure that this footnote is entirely necessary as its value to the argument is not evident.
4. Page 8, lines 343-345: To refer to pages of journals or books as 2D is bordering on tautology, perhaps “2D pages” can be changed simply to “pages”.
5. Page 13, lines 479-480: Shouldn’t the sentence starting “Now, let me analyze” be a new paragraph and not a bullet point?
Author Response
Reviewer 2.
This article revisits the phase rule of Gibbs from the 1870s. Compelling arguments are presented with regard to the selection of independent components, independent state variables, and degrees of freedom. In addition, the author presents an interesting take on the relationship between the number of coexisting phases and the dimensions of the phase diagram of interest. In my opinion, this paper is rather compelling.
I have a couple of general comments and questions related to this work:
Comment 2.1. Can the author perhaps apply their approach to the system mentioned here: https://doi.org/10.1103/PhysRevLett.125.127803 as further evidence of its robustness?
Reply 2.1. Yes, certainly, thanks a lot for this comment. At least, if we suppose that the critical length of the colloidal particles is in the nano-scale, and so this length influences the molar Gibbs energy of the system. I added this to the paper: “Let us note that the phase rule is similarly altered in colloid systems with nano-sized precipitates, when the size (specific surface area) of the nano-particles act as new non-compositional independent state variable [190-191].“
Comment 2.2. Can the author comment on the implementation of their propositions in machine learning systems [e.g. https://doi.org/10.1016/j.scriptamat.2021.114335]. I suppose implementation would be straightforward; are there any plans to undertake such work in future?
Reply 2.2. Thanks again. Yes, I think the extended phase rule as suggested in this paper should be used also in machine learning. As I am not active in machine learning, I only hope that machine-learning-people will read this paper and use the extended phase rule in their further papers. The following sentence is added: “Application of machine learning in constructing phase diagrams is an important new sub-field. However, for higher efficiency those algorithms should be coupled with the phase rule [231]. The future machine learning algorithms should be coupled with the extended phase rule as shown in this paper.“
Here are few minor comments or questions related to the manuscript:
Comment 2.3. Page 6, lines 264 & footnote: I’m not sure that this footnote is entirely necessary as its value to the argument is not evident.
Reply 2.3. Thank you for this. You are right, the footnote is removed, sorry for being super-enthusiastic about the power of mathematics in materials.
Comment 2.4. Page 8, lines 343-345: To refer to pages of journals or books as 2D is bordering on tautology, perhaps “2D pages” can be changed simply to “pages”.
Reply 2.4. Yes, I agree that the claim that pages are 2D pages is obvious. On the other hand, I think it helps understanding why we prefer 2D diagrams with two axes with 2 independent state variables along the axes. So, I made a compromise here and removed some of the “2D”-s, but kept some others.
Comment 2.5. Page 13, lines 479-480: Shouldn’t the sentence starting “Now, let me analyze” be a new paragraph and not a bullet point?
Reply 2.5. Yes, certainly, this was my intention, but it was confused during formatting, sorry for that. I hope it is visible correctly now (although the formatting comes later).
Reviewer 3 Report
Comments and Suggestions for Authors
- Materials engineers often use the phase diagrams with mass fractions. Maybe it should be emphasized in this paper too (page 2, line 73).
- Page 5, line 236. There is no quotation marks in this part of text to which symbol "..." can be referred.
- Page 6, lines 269-271. In fact the P, Pmax, C, ZNc etc. are rather integer numbers. Therefore, the x+y=3 is obeyed by an infinite number of solutions only in general mathematical formulation.
- Fig. 2. - Could you improve the solidus and liquidus lines of gamma phase (please eliminate the bends)
- Reference 70. Please insert space between G. and Kaptay.
Author Response
Reviewer 3
Comment 3.1. Materials engineers often use the phase diagrams with mass fractions. Maybe it should be emphasized in this paper too (page 2, line 73).
Reply 3.1. Thanks for this, this comment is included into the paper (see before Fig.1).
Comment 3.2: Page 5, line 236. There is no quotation marks in this part of text to which symbol "..." can be referred.
Reply 3.2. Sorry, I meant here thew 3 points in , but now it is re-written as “where the three points mean.”
Comment 3.3. Page 6, lines 269-271. In fact the P, Pmax, C, ZNc etc. are rather integer numbers. Therefore, the x+y=3 is obeyed by an infinite number of solutions only in general mathematical formulation.
Reply 3.3. Yes, but in the phase rule the equations are meant the equalities of partial molar Gibbs energies of the same component in different phases with non-integer unknowns, such as mole fractions.
Comment 3.4. Fig. 2. - Could you improve the solidus and liquidus lines of gamma phase (please eliminate the bends)
Reply 3.4. Sorry for this. Now, I made my best to make both the solidus and liquidus lines continuous, and hopefully now the figure looks better.
Comment 3.5. Reference 70. Please insert space between G. and Kaptay.
Reply 3.5. Thanks for this, it is corrected.
Reviewer 4 Report
Comments and Suggestions for Authors The author of this manuscript made a very complete review of the Gibbs phase rule. The review includes numerous cases that are corroborated with pertinent examples. It's an excellent tribute to 50 years of Gibbs' phase ruleAuthor Response
Reviewer 4.
Comment 4.1. The author of this manuscript made a very complete review of the Gibbs phase rule. The review includes numerous cases that are corroborated with pertinent examples. It's an excellent tribute to 50 years of Gibbs' phase rule
Reply 4.1. Thanks a lot.
Round 2
Reviewer 1 Report
Comments and Suggestions for Authors
I see the manuscript has been extended and improved in certain aspects, having satisfied or clarified some of the points I raised in my previous revision. However, I still see some important issues to resolve.
First of all, the “Example 3 for chemical engineers” (at the end of section 2) is not clear enough and does not seem to be appropriate. CH, CH2 and CH3 can be radicals or possible groups to build a molecule in a group-contribution model like UNIFAC, but are not stable compounds or molecules. Moreover, in Reply 1.3 the author expresses that “C” should be counted as the number of elements. In Chemical Engineering, as in many other areas, compounds, components and species are normally used as synonyms, meaning molecules. CH4 is the only molecule among the species considered in the example. Other molecules could be ethane (C2H6), n-eicosane (C20H42), benzene (C20H42), water, carbon dioxide, etc.
When applying the phase rule, if we consider a mixture of 5 hydrocarbons (like methane, n-eicosane, benzene and others) C is definitely 5 and not 2!!! Accordingly, in a mixture of methane and carbon dioxide, C is clearly 2, not 3.
Of course, this implies assuming that the system is not reactive, which is realistic, valid and useful in most situations. Otherwise, we could not treat binary or ternary systems, or multicomponents mixtures, as we do. See for example chapter 16 in the book of Richard Elliott “Introductory Chemical Engineering Thermodynamics” or any other Thermodynamics book in the area of Chemical, Food or Petroleum Engineering, etc
I especially recommend Prof. Kaptay to see Chapter 9 in the book by O’Connell and Haile, where the phase rule is applied to different types of systems and situations:
J.P. O´Connell, J.M. Haile, Thermodynamics. Fundamentals for Applications, First edit, Cambridge University Press, New York, USA, 2005.
Even for reacting systems, they introduce the generalized phase rule in section 10.3, in particular Eq. 10.3.1: F = C + 2 – P – R – S
where the number of independent reactions (R) is subtracted to the classical form of the phase rule for non-reacting systems. C continues to be the number of molecules or compounds, not elements.
Therefore, coming back to the manuscript being considered for publication, I think the author should also consider this type of approach, which may be different to the one typically followed in Metallurgy, but is the standard one in many other areas and fields of application.
Finally, in Reply 1.3 the author also claimed that “the phase rule of Gibbs does not consider kinetic constraints, slowing down the reaction…” meaning that it could not be applied to what we normally consider non-reactive systems. I am not aware of that type of restriction in the original works by Gibbs. If there is something like that, or supporting the idea that C should be equaled to the number of elements instead of compounds/molecules, the author should quote that and provide the specific reference.
Regarding Reply 1.6, the author wrote in the revised manuscript: “in real life there is a negligible probability to find the exact special pressure value (at fixed temperature) or the exact special temperature value (at fixed pressure) at which 4 phases indeed coexist in equilibrium in a ternary system.” From that, it could be understood that 4-phase equilibrium is not observable in practice for a ternary system. I completely disagree, and I keep on my simple reasoning and argument, which I will now extend more. Although much less frequent and registered, 4-phase equilibrium in a ternary system is like 3-phase equilibrium in a binary system, which is extensively documented. Each of those situations has one degree of freedom. If we design an isothermal experiment for any of those situations, where pressure is gradually increased, very slowly so the system reaches equilibrium after each increment, then (if the mixture composition and the temperature are appropriate) we will observe the transformation from one biphasic to another biphasic separation in a ternary mixture, before and after the pressure of the 3-phase equilibrium (or before and after the corresponding temperature in an isobaric diagram, as in Fig. 3 in this manuscript). The transition will imply 3-phase equilibrium and, necessarily, it will be observable during some period of time, beginning with the appearance of the first drop of the new phase and finishing with the disappearance/dissolution of the last drop/bubble/particle of the phase that will not take part in the second situation. Since there is some volume change and some enthalpy change associated to that process, it cannot occur instantaneously. Then, the equilibrium is observable. The same applies to 4-phase equilibrium in a ternary system. Indeed, there is an interesting Review by Adrian et al. in The Journal of Supercritical Fluids 12 (1998) 185–221, “High-pressure multiphase behaviour of ternary systems carbon dioxide–water–polar solvent: review and modeling with the Peng–Robinson equation of state”. Table 1 in that work (which is already more than 25 years old) reported several references where 4-phase equilibrium was observed, measured and reported for ternary systems.
In view of Reply 1.8, the author should declare explicitly in the paper that specification of variables like overall enthalpy or volume would allow to calculate phase fractions in those cases, but they are not considered in this work.
In relation to Reply 1.9, the variables that should be considered in the phase rule are not necessarily the ones for which “engineers have technical ways to define”, but the intensive variables that define the equilibrium state of the system. Average mole fractions are not that type of variables, since they do not affect the fugacities or chemical potentials involved in the equilibrium conditions (Eqs 3 or 4 in the manuscript) and this can also be seen from the fact that, in the example discussed of a binary system at given values of T and P, the overall composition can be changed without affecting the equilibrium state, only the phase fraction. Instead, either T, P or the components mole fractions in each phase, each of them do affect the equilibrium state.
Even the author himself, in the derivation of the phase rule in section 4, acknowledges that “For each phase 𝛷 there are 𝐶−1 independent mole fractions 𝑥i(Φ)…” So, why does he insist in considering the overall mole fractions instead? The degrees of freedom come, as in other similar problems, from subtracting the number of equations to the number of independent variables involved in those equilibrium equations.
If that point is clarified (as in many textbooks like the one by O’Connell and Haile, previously recommended), then there is no need for obscure discussions like the one in section 10 (it is not even clear what the author means by “the selected value of the X physical quantity.”) or strange definitions in section 3.
Author Response
Dear Editor and Reviewer,
I thank the reviewer for his/her further questions and remarks. Here I provide replies to all of them. All yellow marks were removed from the R1 version of the paper and new yellow marks are added in this R2 version of the paper. I hope the paper will be accepted in its revised form.
Sincerely, George Kaptay, sole author
General comment: I see the manuscript has been extended and improved in certain aspects, having satisfied or clarified some of the points I raised in my previous revision. However, I still see some important issues to resolve.
Comment R2.1 (regarding Reply R1.1.3).
Formal reply R2.1. Let me divide this long comment into paragraphs as parts A-E, so in this way it is easier to reply.
Comment R2.1A. First of all, the “Example 3 for chemical engineers” (at the end of section 2) is not clear enough and does not seem to be appropriate. CH, CH2 and CH3 can be radicals or possible groups to build a molecule in a group-contribution model like UNIFAC, but are not stable compounds or molecules. Moreover, in Reply 1.3 the author expresses that “C” should be counted as the number of elements. In Chemical Engineering, as in many other areas, compounds, components and species are normally used as synonyms, meaning molecules. CH4 is the only molecule among the species considered in the example. Other molecules could be ethane (C2H6), n-eicosane (C20H42), benzene (C20H42), water, carbon dioxide, etc.
Reply R2.1A. Thanks for this. This paragraph contains three sub-comments, let me reply one by one. Regarding Example 3, I just picked the simplest formula units for my example, but the same can be repeated using more complex formula units. So, Example 3 is re-written as: “Example 3 for chemical engineers: suppose the species of interest are methane (CH4 with 0.800 mole fraction of H), ethane (C2H6, with 0.750 mole fraction of H) and benzene (C20H42 with 0.678 mole fraction of H). Although there are three species, there are only two elements (C and H), so maximum two components can be selected. One of the possibilities is to select carbon and H2, as reacting them with each other all the three species of interest can be created (at least in principle). But this is not a usual choice of a chemist. For a chemist an ideal choice for the two components is CH4 and C20H42, as combining them in a given ratio the third specie of interest can be created (at least in principle). This is because the mole fraction range of H in these two components (0.678 … 0.800) covers the mole fractions of H in C2H6 (0.750). Note that the choice of CH4 and C2H6 (as the mixture of the two simplest species from the above list) as two components would be wrong, as the third specie of interest cannot be created from their combination, even in principle. It is because the mole fraction range of H in these two components (0.750 … 0.800) excludes the mole fractions of H in C20H42 (0.678).“. Second, please, note that I do not claim in the component rule that C equals the number of elements, I rather claim that the maximum possible value for C is the number of elements in all species considered, see text: “the maximum number of components to be selected for phase equilibria calculations from a list of species of interest (elements and compounds) equals the total number of elements contained in all those species of interest.”. And third, yes, I agree: it is unfortunate that in chemical engineering and in other areas (including metallurgy) compounds, components, species and molecules (and also atoms and elements) are used as synonyms. This simplified view leads to many problems. The following text is added to clarify this point: “Unfortunately, in some fields and in some simplified texts expressions like elements, atoms, molecules, compounds, species and components are used as synonyms. So, before the number of components is defined here, let me show the differences between the meanings of these words.
First, let me explain the difference between elements and atoms. Atoms exist in Nature and their existence has been proven at least for a century. Although we know that atoms are made of elementary particles, nevertheless in materials science atoms are convenient to consider as the smallest building blocks of materials (sometimes remembering how important are electrons in making chemical bonds, or in electrochemistry). For example, silicon atoms can diffuse and evaporate, can be part of a metallic alloy or can make chemical bonds with other atoms. On the other hand, the word “element” has an abstract meaning: silicon, as an element and its symbol Si can be positioned into the Periodic System and it can be discussed. However, silicon as an element does not diffuse.
Molecules are made of atoms. Molecules are real entities, they can diffuse, dissociate, evaporate, etc. On the other hand, the word “compound” has an abstract meaning. We can talk about compounds, but they do not diffuse.
Both words species and components have abstract meaning. The word species is used in the most general sense: they are all entities that can be described by a chemical formula unit, such as C, H, H2, CH, CH4, etc… That is why in this paper I discuss “species of interest” to the engineer and to the researcher, meaning that any specie can come to a mind of an engineer and he/she can be interested in any specie. However, the word component is used in a narrower sense here: all components are species, but not all species are necessarily components. Components in this paper are treated in thermodynamic sense: components are those chemically independent selected species, which determine phase equilibria in a given system. That is why components play a crucial role in this paper, as the phase rule is about phase equilibria.”
Comment R2.1B.When applying the phase rule, if we consider a mixture of 5 hydrocarbons (like methane, n-eicosane, benzene and others) C is definitely 5 and not 2!!! Accordingly, in a mixture of methane and carbon dioxide, C is clearly 2, not 3.
Reply R2.1B. This paragraph is made of two parts. In the first part (“C is 5 and not 2”) you mean the case when the 5 hydrocarbons do not dissolve in each other and do not react. For the case of non-reactive systems see my reply R2.1E below. However, if they dissolve each other, then it is sufficient to select two components (see Example 3 above) for phase equilibria calculations. And yes, in this case 3 species are neglected in phase equilibria calculations and their equilibrium compositions might be of interest. Let me remind the following sentence in the paper: “Following the above rules, usually some species of interest are excluded from the list of components. However, after phase equilibrium is found as step 1 (discussed here), the concentrations of all the species of interest can be calculated in all equilibrium phases as step 2, using the widely known methods of thermodynamics of chemical reactions.” In the second part (“C = 2 not 3”) you are right: you can select the number of components being below the maximum number, as written in the paper: “To simplify the situation also a smaller number of components can be selected than the maximum value given by the component rule, supposing the above rules are obeyed. However, in this case usually the dissociation of compound-components cannot be studied.“ Also please note that the component rule is only about the maximum number of components, as written in the paper: “the maximum number of components to be selected for phase equilibria calculations from a list of species of interest (elements and compounds) equals the total number of elements contained in all those species of interest.”.
Comment R2.1C. Of course, this implies assuming that the system is not reactive, which is realistic, valid and useful in most situations. Otherwise, we could not treat binary or ternary systems, or multicomponents mixtures, as we do. See for example chapter 16 in the book of Richard Elliott “Introductory Chemical Engineering Thermodynamics” or any other Thermodynamics book in the area of Chemical, Food or Petroleum Engineering, etc
Reply R2.1C. Yes, I agree it is a useful concept. But when this concept is applied, the phase rule of Gibbs is not valid – see reply R2.1E below.
Comment R2.1D. I especially recommend Prof. Kaptay to see Chapter 9 in the book by O’Connell and Haile, where the phase rule is applied to different types of systems and situations: J.P. O´Connell, J.M. Haile, Thermodynamics. Fundamentals for Applications, First edit, Cambridge University Press, New York, USA, 2005. Even for reacting systems, they introduce the generalized phase rule in section 10.3, in particular Eq.(10.3.1): F = C + 2 – P – R – S, where the number of independent reactions (R) is subtracted to the classical form of the phase rule for non-reacting systems. C continues to be the number of molecules or compounds, not elements. Therefore, coming back to the manuscript being considered for publication, I think the author should also consider this type of approach, which may be different to the one typically followed in Metallurgy, but is the standard one in many other areas and fields of application.
Reply R2.1D. Thank you for this. The equation F = C + 2 – P – R is used in my paper by reducing the number of chemically independent components as (C – R) at the very beginning, so R (the number of independent chemical reactions) do not complicate the further equations. Please, see my sentence in the paper with the same meaning: “Sometimes the number of independent components is calculated as the number of species minus the number of independent reactions between them [3-4]. However, if the number of species is large (as for example in organic chemistry or polymers engineering), this is not an easy task.”. That is why the component rule is introduced in this paper, which has the same sense, but it is easier to apply. A new sentence is added to make it more clear: “Let me note that the component rule is identical with the idea written above: the number of independent components is the number of species minus the number of independent chemical reaction between them. However, it is easier to count the number of elements in the mixture of 100 hydrocarbons (= 2: C and H) than counting the number of independent chemical reactions between them (= 98).”
Comment R2.1E. Finally, in Reply 1.3 the author also claimed that “the phase rule of Gibbs does not consider kinetic constraints, slowing down the reaction…” meaning that it could not be applied to what we normally consider non-reactive systems. I am not aware of that type of restriction in the original works by Gibbs. If there is something like that, or supporting the idea that C should be equaled to the number of elements instead of compounds/molecules, the author should quote that and provide the specific reference.
Reply R2.1E. Regarding the first part of the question on kinetic constraints, Gibbs wrote in his chapter entitled “On coexistent phases of matter” (see pp.96-100 in “The Collected Works of J Willard Gibbs in 2 volumes. volume I. Thermodynamics. Longmans, Green and Co, NY – London-Toronto, 1928., pp. 55-353): “We may call such bodies as differ in composition or state different phases of the matter considered, regarding all bodies which differ only in quantity and form as different examples of the same phase. Phases which can coexist together, the dividing surfaces being plane, in an equilibrium which does not depend upon passive resistances to change, we shall call coexistent phases.“. Based on this, I wrote in my paper: The phase rule of Gibbs is about the number of equilibrium phases in the system (P). Phases are 3D bodies that differ from each other by their inner structure and/or composition. If the same phase is broken into many parts (as many bubbles, droplets or crystals) it does not increase the number of phases, i.e. 1 million graphite crystals in the same crucible is one phase. The phase rule of Gibbs connects the maximum number of phases (Pmax) that can coexist with planar interfaces in an equilibrium with the number of independent components in the system (C) [1]:
(1a)
where “2” corresponds to pressure and temperature. It is important to note that both the original phase rule of Gibbs and its extension offered here are valid only in truly equilibrium systems, in which chemical reactions and phase changes are not limited by kinetic constraints. In real life, however, especially at relatively low temperatures “non-reacting systems” are sometimes discussed, in which chemical reactions and/or phase changes seem not happen due to our limited human life-time. The phase rule of Gibbs should not be applied to such “non-reactive systems”, simply because those are not in real equilibrium. In fact, in those non-equilibrium, non-reacting systems one can find as many phases as many are added by the engineer to the system, in seeming contradiction with Eq.(1a).” In my paper I wrote “not limited by kinetic constraints”, which I think is a shorter equivalent to the wording by Gibbs “which does not depend upon passive resistances to change”. To my understanding it means that the phase rule of Gibbs relates to systems which are in full chemical and phase equilibrium. In other words, so called “non-reactive systems” cannot be discussed using the phase rule. Relating to the second part of your question on the number of elements I do not claim in my paper that Gibbs wrote it. In contrary, I claim that this is one of the novelties of my paper discussed (that is why the component rule is given even in the title of my paper). This is because the work of Gibbs is quite general, he did not care too much about differences between elements, atoms and molecules. Especially that their existence was not really proven in 1870s.
Comment R2.2. (regarding Reply R1.1.6). Regarding Reply 1.6, the author wrote in the revised manuscript: “in real life there is a negligible probability to find the exact special pressure value (at fixed temperature) or the exact special temperature value (at fixed pressure) at which 4 phases indeed coexist in equilibrium in a ternary system.” From that, it could be understood that 4-phase equilibrium is not observable in practice for a ternary system. I completely disagree, and I keep on my simple reasoning and argument, which I will now extend more. Although much less frequent and registered, 4-phase equilibrium in a ternary system is like 3-phase equilibrium in a binary system, which is extensively documented. Each of those situations has one degree of freedom. If we design an isothermal experiment for any of those situations, where pressure is gradually increased, very slowly so the system reaches equilibrium after each increment, then (if the mixture composition and the temperature are appropriate) we will observe the transformation from one biphasic to another biphasic separation in a ternary mixture, before and after the pressure of the 3-phase equilibrium (or before and after the corresponding temperature in an isobaric diagram, as in Fig. 3 in this manuscript). The transition will imply 3-phase equilibrium and, necessarily, it will be observable during some period of time, beginning with the appearance of the first drop of the new phase and finishing with the disappearance/dissolution of the last drop/bubble/particle of the phase that will not take part in the second situation. Since there is some volume change and some enthalpy change associated to that process, it cannot occur instantaneously. Then, the equilibrium is observable. The same applies to 4-phase equilibrium in a ternary system. Indeed, there is an interesting Review by Adrian et al. in The Journal of Supercritical Fluids 12 (1998) 185–221, “High-pressure multiphase behaviour of ternary systems carbon dioxide–water–polar solvent: review and modeling with the Peng–Robinson equation of state”. Table 1 in that work (which is already more than 25 years old) reported several references where 4-phase equilibrium was observed, measured and reported for ternary systems.
Reply R2.2. In the 3-component phase diagram with one non-compositional state variable (p with T = constant or T with p = constant): Pmax = 3 + 1 = 4. So, a 4-phase point exists in such phase diagrams if the 3D diagram is plotted as function of compositions + pressure. However, experiments are run with fixed p and fixed T. So, at fixed T you should find the special p value that corresponds to this 4-phase equilibrium, such as p = 1.2345678901234567890 bar (the number of digits can be increased to infinity). Now, what is the probability you can set this pressure exactly in a system? I think it is nil. You can observe four phases only for kinetic reasons. When you change the pressure from p1 (with one 3-phase equilibrium) to p2 (with another 3-phase equilibrium), temporarily you can observe 4 phases, but they are not in equilibrium. One phase is in the process of being disappeared, while another phase is being appeared. However, both processes take some time in non-equilibrium state, so there is some time period when you observe 4 phases together. This part is re-written as: “Note also, that by increasing temperature a special temperature can be found when the 3-phase region in the middle of Fig.4 is concentrated into a single point, at which also a liquid phase appears: this point is called the ternary eutectic point with 4 co-existing phases (α+β+γ+l). This point can be shown in a 3D phase diagram as function of concentrations and temperature (at fixed pressure). Similarly, a 4-phase point can be shown as function of compositions and pressure (at fixed temperature) in a 3D phase diagram [30]. The horizontal cross section at this ternary eutectic temperature is not shown in Fig.4, as the probability of finding this temperature is practically nil, i.e. 4 phases cannot be experimentally found to coexist in ternary systems with fixed pressure and temperature values. This also follows from Table 1, predicting F = -1 for this case, further proving that this case does not exist, in agreement with Eq.(6b) shown above. In the same way we can conclude that 4 phases cannot coexist in binary systems with fixed pressure and variable temperature or fixed temperature and variable pressure. For the same reason 4 phases cannot coexist in equilibrium in one-component systems, even if both pressure and temperature are variables. This is because in all these cases Pmax = 3 and F = -1 for the case of P = 4. This argument was written by Gibbs as: “it is entirely improbable that there are four coexistent phases of any simple substance” [1].“ Let me note that in this point my paper is not different from the paper of Gibbs.
Comment R2.3. (regarding Reply R1.1.8). In view of Reply 1.8, the author should declare explicitly in the paper that specification of variables like overall enthalpy or volume would allow to calculate phase fractions in those cases, but they are not considered in this work.
Reply R2.3. As per your request, a new sentence is added as: “Let me also note that the phase fractions would be calculable from measured enthalpy or volume of the system, but they are not considered in this paper as they are not independent state variables.“
Comment R2.4. (regarding reply R1.1.9). In relation to Reply 1.9, the variables that should be considered in the phase rule are not necessarily the ones for which “engineers have technical ways to define”, but the intensive variables that define the equilibrium state of the system. Average mole fractions are not that type of variables, since they do not affect the fugacities or chemical potentials involved in the equilibrium conditions (Eqs 3 or 4 in the manuscript) and this can also be seen from the fact that, in the example discussed of a binary system at given values of T and P, the overall composition can be changed without affecting the equilibrium state, only the phase fraction. Instead, either T, P or the components mole fractions in each phase, each of them do affect the equilibrium state. Even the author himself, in the derivation of the phase rule in section 4, acknowledges that “For each phase ? there are ?−1 independent mole fractions ?i(Φ)…” So, why does he insist in considering the overall mole fractions instead? The degrees of freedom come, as in other similar problems, from subtracting the number of equations to the number of independent variables involved in those equilibrium equations. If that point is clarified (as in many textbooks like the one by O’Connell and Haile, previously recommended), then there is no need for obscure discussions like the one in section 10 (it is not even clear what the author means by “the selected value of the X physical quantity.”) or strange definitions in section 3.
Reply R2.4A. Suppose for a moment that you are right and the state variables are p + T + the mole fractions in each phase. Then, for a binary, 2-phase system with fixed value of p we have 3 state variables: T, xB(alpha) and xB(beta). If so, the phase diagram at p = const should be shown in 3D, as function of these 3 independent state variables. But this is never the case. The binary phase diagrams are in fact shown at fixed p as 2D figures, as function of T and xB (average composition of the system) – see Fig.3 and thousands (millions?) of actual binary phase diagrams in handbooks and in internet. Indeed, the molar Gibbs energies of components in phase beta are functions of p-T-xB(beta) and the molar Gibbs energies of components in phase alpha are functions of p-T-xB(alpha). This is the way how the unknowns xB(alpha) and xB(alpha) are found solving Eq-s (3-4) as function of T and xB (and p). See also Fig.1 and text before. Once these originally unknown xB(alpha) and xB(beta) equilibrium values are found, then also activities and fugacities of A and B components in both phases follow at given T and p.
